# Peripheral opioid receptor antagonism alleviates fentanyl-induced cardiorespiratory depression and is devoid of aversive behavior

Brian C Ruyle[1,2,3], Sarah Masud[1,2,3], Rohith Kesaraju[1,2,3], Mubariz Tahirkheli[1,2,3], Juhi Modh[1,2,3], Caroline G Roth[1,2,3], Sofia Angulo-Lopera[1,2,3], Tania Lintz[1,2,3], Jessica A Higginbotham[1,2,3], Nicolas Massaly[1,2,3], Jose A Morón[1,2,3,4,5]*

[1]Department of Anesthesiology, Washington University in St. Louis, St. Louis, United States; [2]Pain Center, Washington University in St. Louis, St. Louis, United States; [3]School of Medicine, Washington University in St. Louis, St. Louis, United States; [4]Department of Neuroscience, Washington University in St. Louis, St. Louis, United States; [5]Department of Psychiatry, Washington University in St. Louis, St. Louis, United States

*For correspondence:
jmoron-concepcion@wustl.edu

Competing interest: The authors declare that no competing interests exist.

## eLife Assessment

This manuscript represents a **fundamental** contribution demonstrating that fentanyl-induced respiratory depression can be reversed with a peripherally-restricted mu opioid receptor antagonist. The paper reports **compelling** and rigorous physiological, pharmacokinetic, and behavioral evidence supporting this major claim, and furthers mechanistic understanding of how peripheral opioid receptors contribute to respiratory depression. These findings reshape our understanding of opioid-related effects on respiration and have significant therapeutic implications given that medications currently used to reverse opioid overdose (such as naloxone) produce severe aversive and withdrawal effects via actions within the central nervous system.

**Abstract** Millions of Americans suffering from Opioid Use Disorders face a high risk of fatal overdose due to opioid-induced respiratory depression (OIRD). Fentanyl, a powerful synthetic opioid, is a major contributor to the rising rates of overdose deaths. Reversing fentanyl overdoses has proved challenging due to its high potency and the rapid onset of OIRD. We assessed the contributions of central and peripheral mu opioid receptors (MORs) in mediating fentanyl-induced physiological responses. The peripherally restricted MOR antagonist naloxone methiodide (NLXM) both prevented and reversed OIRD to a degree comparable to that of naloxone (NLX), indicating substantial involvement of peripheral MORs to OIRD. Interestingly, NLXM-mediated OIRD reversal did not produce aversive behaviors observed after NLX. We show that neurons in the nucleus of the solitary tract (nTS), the first central synapse of peripheral afferents, exhibit a biphasic activity profile following fentanyl exposure. NLXM pretreatment attenuates this activity, suggesting that these responses are mediated by peripheral MORs. Together, these findings establish a critical role for peripheral MORs, including ascending inputs to the nTS, as sites of dysfunction during OIRD. Furthermore, selective peripheral MOR antagonism could be a promising therapeutic strategy for managing OIRD by sparing CNS-driven acute opioid-associated withdrawal and aversion observed after NLX.

## Introduction

Millions of Americans live with Opioid Use Disorders (OUD) and face a high risk of opioid-induced respiratory depression (OIRD), the leading cause of opioid related deaths (*Ramirez et al., 2021*). The number of fatalities attributed to OIRD has been exacerbated by the rise of distribution and use of synthetic opioids, such as fentanyl (*Comer and Cahill, 2019*; *Mattson et al., 2021*; *Parida et al., 2019*; *Wilson et al., 2020*). Fentanyl is a highly lipophilic opioid that readily crosses the blood–brain barrier and binds tightly to mu opioid receptors (MORs), which are abundant in the respiratory centers of the brainstem (*Ramirez et al., 2021*; *Mansour et al., 1994*; *Zhuang et al., 2017*). Compared to other opioids such as heroin and morphine, fentanyl exhibits both a faster onset of OIRD and higher potency for MOR binding (*Hill et al., 2020*; *Marchette et al., 2023*), minimizing successful prevention of lethal outcomes using naloxone (NLX), a competitive and preferential MOR antagonist (*Lewanowitsch and Irvine, 2003*; *Fairbairn et al., 2017*). Despite its high efficacy in reversing OIRD, NLX also precipitates unpleasant withdrawal symptoms and aversion, as reported in patients and preclinical models (*Fairbairn et al., 2017*; *Lewanowitsch and Irvine, 2002*; *Lai et al., 2021*; *Yugar et al., 2023*), making its implementation for managing OUD challenging. Therefore, a better understanding of the pharmacology, specific brain regions and pathways, and physiological responses induced by high doses of fentanyl are necessary to develop effective prevention and intervention strategies, ultimately saving lives of patients suffering from OUD.

Several regions have been implicated as key sites of triggering OIRD, including the Pre-Botzinger Complex in the ventral medulla and the parabrachial nucleus and Kölliker–Fuse nucleus in the pons (*Ramirez et al., 2021*; *Bachmutsky et al., 2020*; *Bateman and Levitt, 2023*; *Haji et al., 2003*; *Levitt et al., 2015*; *Palkovic et al., 2020*; *Montandon et al., 2011*). Opioids act directly within these networks to alter inspiratory and expiratory phase duration, leading to overall reductions in respiratory rate and apneas (*Levitt et al., 2015*; *Montandon et al., 2011*; *Pattinson, 2008*). In addition, the nucleus of the solitary tract (nTS), located in the dorsal brainstem, is the first central site that receives sensory afferent information via cranial nerves related to cardiorespiratory, gustatory, and gastrointestinal function (*Andresen and Kunze, 1994*). Acute hypoxia robustly activates nTS neurons (*King et al., 2012*; *Teppema et al., 1997*) to initiate appropriate autonomic and cardiorespiratory responses that facilitate a return to homeostatic physiological states. MORs are expressed throughout the entire caudal–rostral extent of the nTS, where they are located postsynaptically on nTS neurons and on vagal afferent fibers that terminate within the nTS (*Aicher et al., 2000*; *Maletz et al., 2022*; *Furdui et al., 2024*). Opioids induce both systemic and brain hypoxia (*Disney et al., 2022*; *Solis et al., 2018*) and produce an increase in Fos-immunoreactivity (IR) in the nTS (*Maletz et al., 2022*; *Salas et al., 2013*). However, engaging MOR signaling in the nTS impairs hypoxic ventilatory responses (*Zhuang et al., 2017*). This suggests that OIRD may result from nTS dysfunction failing to engage appropriate cardiorespiratory responses, thereby leading to prolonged OIRD. However, the specific contributions of nTS MOR, including ascending MOR-expressing inputs from the periphery to this region, in driving aberrant cardiorespiratory depression are still not fully understood.

In addition to their high expression throughout brainstem respiratory nuclei, MORs are also expressed in the periphery, including sensory ganglia, lung afferents, cardiac tissue, and cranial nerves that terminate within the nTS (*Cabot et al., 1994*; *Li et al., 1996*). Naloxone methiodide (NLXM) is a quaternary derivative of NLX that does not cross the blood–brain barrier at low doses, making it a useful tool to assess the peripheral versus central contribution of MORs in opioid-induced physiology and behavior (*Lewanowitsch and Irvine, 2003*; *Perekopskiy et al., 2020*). Despite its lower binding affinity for MOR (*Lewanowitsch and Irvine, 2003*), NLXM has been shown to reverse opioid-induced hypoventilation and brain hypoxia in rodent models (*Lewanowitsch and Irvine, 2002*; *Perekopskiy et al., 2020*; *Henderson et al., 2013*), suggesting that peripheral opioid receptors may play a greater role in OIRD than previously thought. However, no studies to date have provided a comprehensive assessment of opioid-induced cardiorespiratory depression and systemic hypoxia combined with a real-time assessment of MOR-mediated activity of nTS neurons.

In the present study, we examined the relative contributions of central and peripheral MORs mediating fentanyl-induced depression of cardiorespiratory parameters and investigated potential mechanisms of opioid-induced dysfunction within the nTS. We report that fentanyl-induced cardiorespiratory depression and prolonged systemic hypoxia can be prevented or reversed by NLXM to degree comparable to NLX. As compared to NLX, NLXM-mediated reversal of OIRD did not produce aversive-like

behaviors. Fentanyl-induced robust Fos-IR expression in the nTS in a dose-dependent manner, and this activation can be attributed to a combination of direct effects of opioids acting at MOR located within the nTS and the subsequent hypoxia that develops after fentanyl exposure. Given that the majority of nTS MOR is located on vagal afferent fibers, we utilized various techniques to evaluate opioid-induced activity within the nTS and manipulate nTS MOR signaling during OIRD. Using wireless in vivo fiber photometry, we show that fentanyl evokes a biphasic activity profile in nTS neurons, characterized by a transient increase followed by a prolonged decrease in neuronal excitability. NLXM pretreatment strongly attenuated this biphasic response, indicating that fentanyl-induced nTS activity is influenced by MOR-expressing peripheral afferents. Together, these findings provide novel insights into peripheral mechanisms mediating OIRD and support peripheral MOR antagonism as a promising therapeutic strategy for managing OIRD for patients suffering from OUD while sparing the CNS-driven acute aversive behaviors that are observed with the use of NLX.

## Results

### Intravenous fentanyl induces respiratory depression and activates nTS neurons in a dose-dependent manner

We first examined the effect of intravenous fentanyl to induce cardiorespiratory depression and activate neurons in the nucleus tractus solitarius (nTS). Catheterized rats received intravenous saline (1 ml/kg) or fentanyl at a range of doses (2, 20, or 50 µg/kg; *Figure 1A*), and cardiorespiratory parameters were measured up to 60 min. The first 20 min of these responses are shown in *Figure 1B–D*. As shown in *Supplementary file 1A*, baseline cardiorespiratory parameters were similar between male and female rats that received 20 µg/kg fentanyl. There was a small but significant difference in basal heart rate in female rats that received 50 µg/kg fentanyl (*Supplementary file 1B*). Neither saline nor 2 µg/kg fentanyl evoked a significant change in cardiorespiratory parameters. In comparison, both 20 and 50 µg/kg fentanyl produced rapid decreases in oxygen saturation, heart rate, and respiratory rate, with the largest decrease occurring after 50 µg/kg fentanyl. The nadir (defined as the lowest point measured after fentanyl) and recovery (calculated as 90% of baseline values prior to fentanyl) for each physiological parameter were assessed in all groups (*Figure 1E–J*). There was no significant difference in the nadir or recovery values between saline- and 2 µg/kg fentanyl-treated rats. In comparison, rats given 50 µg/kg fentanyl exhibited a significantly lower nadir in oxygen saturation (*Figure 1E*) and respiratory rate (*Figure 1I*) compared to 20 µg/kg. In addition, the recovery time was significantly longer in rats that received 50 µg/kg fentanyl compared to 20 µg/kg fentanyl (*Figure 1F, J*). The nadir of bradycardia was similar between 20 and 50 µg/kg rats (*Figure 1G*), although the duration of the response was significantly longer in rats that received the highest dose of fentanyl (*Figure 1H*). Overall, we observed no sex differences in the nadir or recovery time in rats that received 20 µg/kg fentanyl. Female rats that received 50 µg/kg fentanyl exhibited a significantly longer recovery of respiratory rate, although oxygen saturation and heart rate recover times were similar to males (*Supplementary file 1C, D*). We next examined nTS activation (Fos-IR) following intravenous fentanyl (*Figure 1K*). A significant increase in Fos-IR was observed after 2 µg/kg fentanyl (*Figure 1L*), a dose that did not induce cardiorespiratory depression. Further dose-dependent increases in Fos-IR were observed after 20 and 50 µg/kg fentanyl. We evaluated a subpopulation of catecholaminergic nTS neurons, which are critical for full expression of hypoxia-evoked cardiorespiratory responses (*King et al., 2012*; *King et al., 2015*) for activation after intravenous fentanyl (*Figure 1M*). These cells exhibited increased Fos-IR after 2 µg/kg fentanyl and were further activated at the highest dose of fentanyl (50 µg/kg).

### Peripheral opioid receptor antagonism prevents fentanyl-induced respiratory depression

The above data provide a comprehensive assessment of the dose-dependent changes in cardiorespiratory parameters and nTS neuronal activation evoked by intravenous fentanyl. To gain insight into the extent to which central and peripheral opioid receptors mediate these effects, separate cohorts of rats received intravenous administration of the centrally and peripherally acting competitive opioid receptor antagonist naloxone (NLX; 1 mg/kg) or the peripherally restricted opioid receptor antagonist naloxone methiodide (NLXM, 1 and 5 mg/kg). Initial biodistribution experiments were performed to

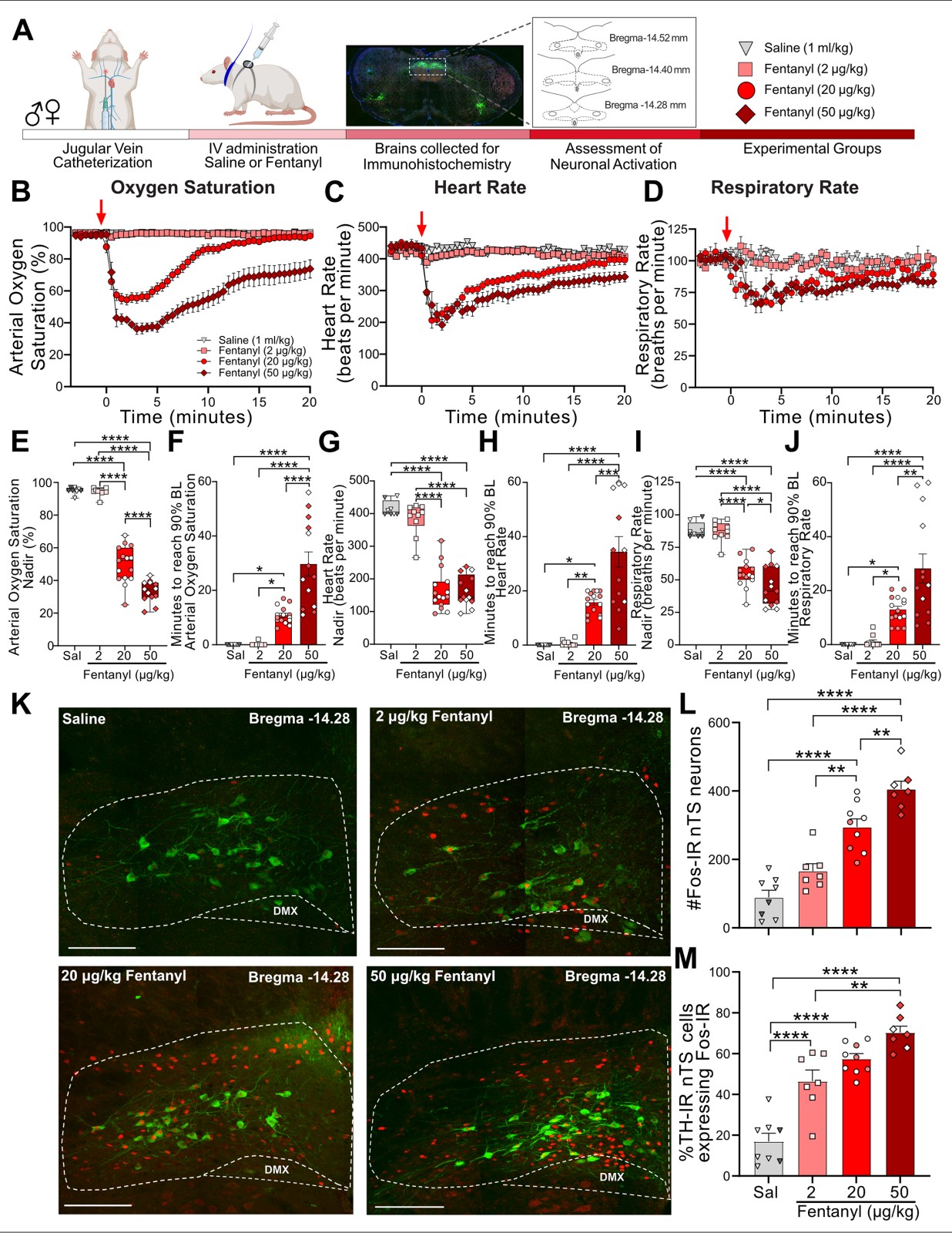

**Figure 1.** Fentanyl induces cardiorespiratory depression and nucleus of the solitary tract (nTS) neuronal activation in a dose-dependent manner. (**A**) Schematic showing the timeline for opioid-induced respiratory depression (OIRD) collar experiments. Male and female SD rats underwent jugular catheterization surgery. One week later, conscious rats in their home cages were fitted with a pulse oximeter collar and received intravenous administration of either saline (*n* = 8, triangles) or fentanyl at doses of 2 µg/kg (*n* = 10, squares), 20 µg/kg (*n* = 15, circles), or 50 µg/kg (*n* = 13,

*Figure 1 continued on next page*

*Figure 1 continued*

diamonds). In all graphs, males are represented by filled symbols, and females by open symbols. Baseline cardiorespiratory parameters were measured for 3 min prior to intravenous administration and up to 60 min after saline or fentanyl. At the conclusion of experiments, brains were collected from a subset of animals in each group and processed for Fos- and TH-immunoreactivity. Time course data showing oxygen saturation (**B**), heart rate (**C**), and respiratory rate (**D**) data measurements in all groups before and after saline or fentanyl. Changes in oxygen saturation were assessed as nadir (lowest value; **E, G, I**) and time to recover to 90% of baseline (pre-saline/fentanyl; **F, H, J**) administration. There was a dose-dependent decrease in the nadir for oxygen saturation (one-way ANOVA, $F_{(3,42)}$ = 186.3, p < 0.0001; Tukey's post hoc ****p < 0.0001), heart rate (one-way ANOVA $F_{(3,42)}$ = 74.90, p < 0.0001; Tukey's post hoc ****p < 0.0001), and respiratory rate (one-way ANOVA, $F_{(3,42)}$ = 45.35, p < 0.0001; Tukey's post hoc ****p < 0.0001, *p < 0.05). Time to recover to 90% baseline oxygen saturation (one-way ANOVA, $F_{(3,42)}$ = 28.72, p < 0.0001; Tukey's post hoc ****p < 0.0001, *p < 0.05), heart rate (one-way ANOVA, $F_{(3,42)}$ = 23.89, p < 0.0001; Tukey's post hoc ****p < 0.0001, ***p < 0.001, **p < 0.01, *p < 0.05) and respiratory rate (one-way ANOVA, $F_{(3,42)}$ = 16.53, p < 0.0001; Tukey's post hoc ****p < 0.0001, **p < 0.01, *p < 0.05). (**K**) Merged photomicrographs of coronal brainstem sections displaying representative Fos- and TH-immunoreactivity in the nTS of rats that received intravenous saline or fentanyl. Scale bar = 200 µm. (**L**) Mean data show that 20 and 50 µg/kg fentanyl induced a significantly greater number of Fos-IR cells in the nTS compared to saline and 2 µg/kg fentanyl (one-way ANOVA, $F_{(3,27)}$ = 37.13, p < 0.0001; Tukey's post hoc ****p < 0.0001, **p < 0.001). (**M**) Mean data showing the percentage of TH-IR nTS neurons expressing Fos-IR. Rats that received 2 µg/kg fentanyl displayed a significantly higher increase in activated TH-IR neurons compared to saline-treated rats. Both 20 and 50 µg/kg fentanyl induced a significantly higher percentage of activation in TH-IR cells compared to saline and 2 µg/kg fentanyl (one-way ANOVA, $F_{(3,27)}$ = 37.13, p < 0.0001; Tukey's post hoc ****p < 0.0001, **p < 0.01).

The online version of this article includes the following source data for figure 1:

**Source data 1.** Excel spreadsheet of statistical source data for *Figure 1*.

confirm the peripheral restriction of NLXM. Catheterized rats that received intravenous NLXM (1 or 5 mg/kg) or NLX (1 mg/kg) were evaluated for detection of NLX or NLXM in brain and plasma samples collected either 2 or 10 min after administration (*Figure 2A*). Neither dose of NLXM was detected within the central nervous system at either time point evaluated (*Figure 2B, C*). Importantly, NLX was not detected in these NLXM-treated groups. In comparison, rats that received intravenous NLX exhibited significant, time-dependent detection of NLX in both plasma and brain samples (*Figure 2D*). Thus our data are in agreement with previous reports confirming the peripheral restriction of NLXM (*Perekopskiy et al., 2020*; *McGaraughty et al., 2005*). We next evaluated cardiorespiratory changes from baseline in rats that received intravenous NLX or NLXM at either dose (*Figure 2—figure supplement 1A–D*). Data indicate that neither NLX nor NLXM significantly altered cardiorespiratory parameters from their baseline parameters (*Figure 2—figure supplement 1E–J*). Based on these findings, we utilized NLXM to further investigate the roles of peripheral MOR antagonism in the context of evaluating OIRD.

Catheterized rats received intravenous NLXM (1 or 5 mg/kg) or NLX (1 mg/kg) prior to intravenous fentanyl administration at 20 µg/kg (*Figure 3A*) or 50 µg/kg (*Figure 3—figure supplement 1A*). As shown in *Figure 3B–D* and *Figure 3—figure supplement 1B–D*, the responses evoked by 20 and 50 µg/kg fentanyl induced an onset and duration of cardiorespiratory depression that was similar to what was observed in *Figure 1*. As expected, NLX pretreatment completely blocked the decrease in cardiorespiratory parameters at both 20 and 50 µg/kg doses of fentanyl. In rats that received 20 µg/kg fentanyl, 1 mg/kg NLXM pretreatment prevented bradycardic and hypoventilatory responses to fentanyl (*Figure 3B–D*). For all parameters, saline-pretreated rats exhibited lower nadir values than NLX- and NLXM-pretreated rats, and there was no significant difference in the nadir or duration between NLX (1 mg/kg) and NLXM (1 mg/kg) pretreated rats that received 20 µg/ kg fentanyl (*Figure 3E–J*). Although there was a small drop in the nadir of oxygen saturation in NLXM-pretreated rats, this was not significantly different than NLX-pretreated animals (*Figure 3E*, p = 0.2292). In rats that received 50 µg/kg fentanyl, 1 mg/kg NLXM pretreatment was unable to prevent the initial decrease in oxygen saturation and heart rate (*Figure 3—figure supplement 1E, G*). However, the duration of these responses was significantly shorter compared to saline-treated rats (*Figure 3—figure supplement 1F, H, J*). A higher dose of NLXM (5 mg/kg), which does not cross the blood–brain barrier (*Figure 2C*), was then used in a separate group of animals that received 50 µg/kg fentanyl. Pretreatment with 5 mg/kg NLXM attenuated cardiorespiratory depression. While the nadir of oxygen saturation was significantly lower than NLX-pretreated rats, the duration of this effect was not significantly different between groups (*Figure 3—figure supplement 1E–J*). No sex differences were observed in the nadir or recovery time of any parameter (*Supplementary file 1C, D*). Together, these data demonstrate that peripheral MOR antagonism can sufficiently prevent OIRD by 20 µg/kg fentanyl and strongly attenuates OIRD evoked by 50 µg/kg fentanyl.

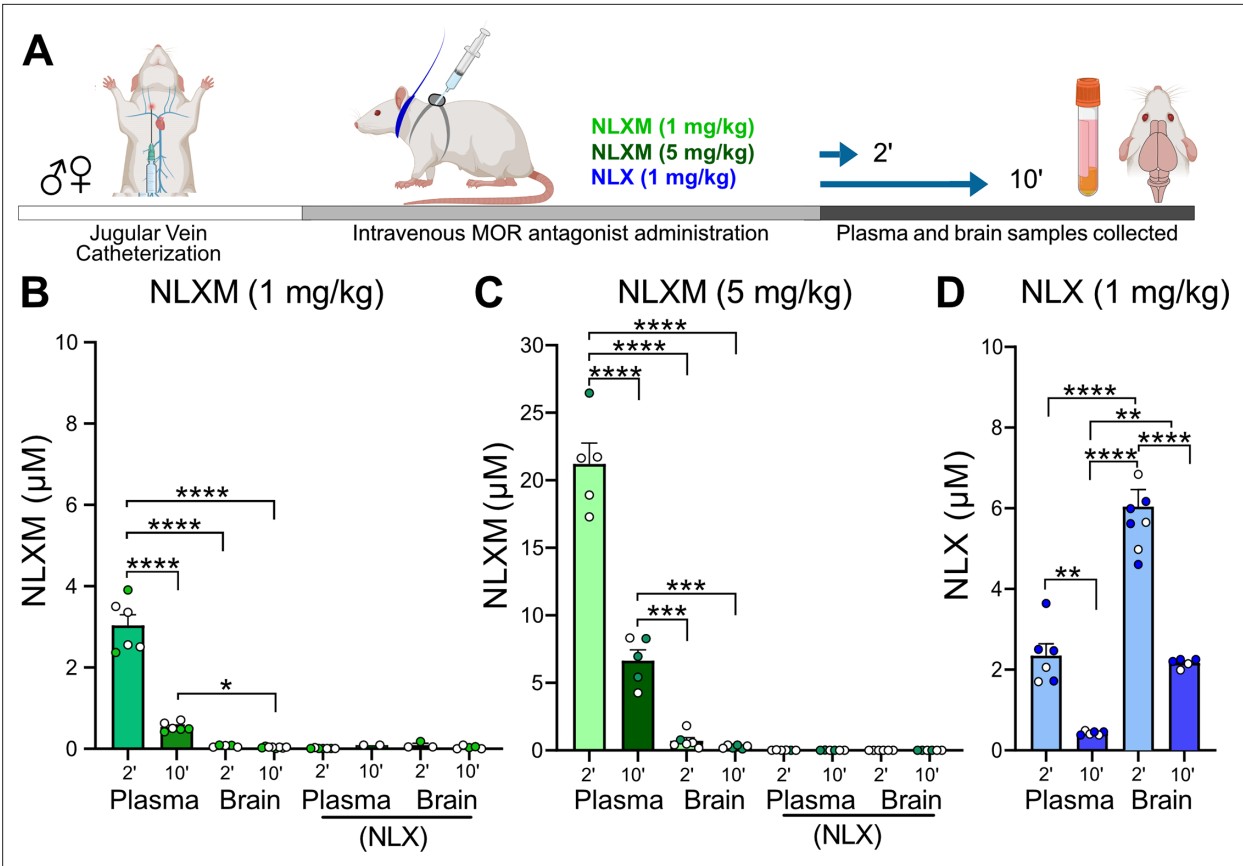

**Figure 2.** NLXM does not cross the blood–brain barrier. (**A**) Timeline of biodistribution experiments. Male and female SD rats underwent intravenous jugular catheterization surgery. The following week, rats were anesthetized and received intravenous NLXM (1 or 5 mg/kg) or NLX (1 mg/kg). Rats were sacrificed 2 and 10 min after intravenous administration of mu opioid receptor (MOR) antagonists. Plasma was extracted from trunk blood and intact brains were removed and rapidly frozen with liquid nitrogen. (**B**) Amount of NLXM detected in plasma and brain samples at each time point after intravenous NLXM (1 mg/kg). One-way ANOVA, $F_{(3,21)}$ = 125.6, p < 0.0001; Tukey's post hoc test, ****p < 0.0001; *p < 0.05. No NLX formation was detected in these samples. (**C**) Amount of NLXM detected in plasma and brain samples at each time point after intravenous NLXM (5 mg/kg). One-way ANOVA, $F_{(3,18)}$ = 144.0, p < 0.0001; Tukey's post hoc test, ****p < 0.0001; ***p < 0.001. No NLX formation was detected in these samples. (**D**) Amount of NLX detected in plasma and brain samples at each time point after intravenous NLX (1 mg/kg). One-way ANOVA, $F_{(3,21)}$ = 66.05, p < 0.0001; Tukey's post hoc test, ****p < 0.0001; **p < 0.01.

The online version of this article includes the following source data and figure supplement(s) for figure 2:

**Source data 1.** Excel spreadsheet of statistical source data for *Figure 2*.

**Figure supplement 1.** Opioid Receptor antagonism has no effect on baseline cardiorespiratory parameters in drug-naïve rats.

**Figure supplement 1—source data 1.** Excel spreadsheet of statistical source data for *Figure 2—figure supplement 1*.

Neuronal activation, as measured by Fos-IR, was examined in the nTS of rats pretreated with saline or opioid receptor antagonists followed by 20 µg/kg (*Figure 3K*) or 50 µg/kg fentanyl (*Figure 3— figure supplement 1K*). Saline pretreatment evoked the highest degree of fentanyl-induced Fos-IR in the nTS, including in a subpopulation of TH-IR neurons. Blocking peripheral MOR signaling with NLXM did not decrease fentanyl-induced nTS neuronal activation, while NLX, which blocks both central and peripheral MORs, resulted in a significant reduction in nTS activation (*Figure 3L, M*). In rats that received the highest dose of fentanyl, the number of nTS cells displaying Fos-IR was similar between rats pretreated with saline and NLXM at both doses, with the only significant reduction occurring in NLX-pretreated animals (*Figure 3—figure supplement 1L*). This degree of activation was also observed in catecholaminergic cells (*Figure 3—figure supplement 1M*), although around 30% of TH-IR neurons also displayed Fos-IR in NLX-pretreated rats.

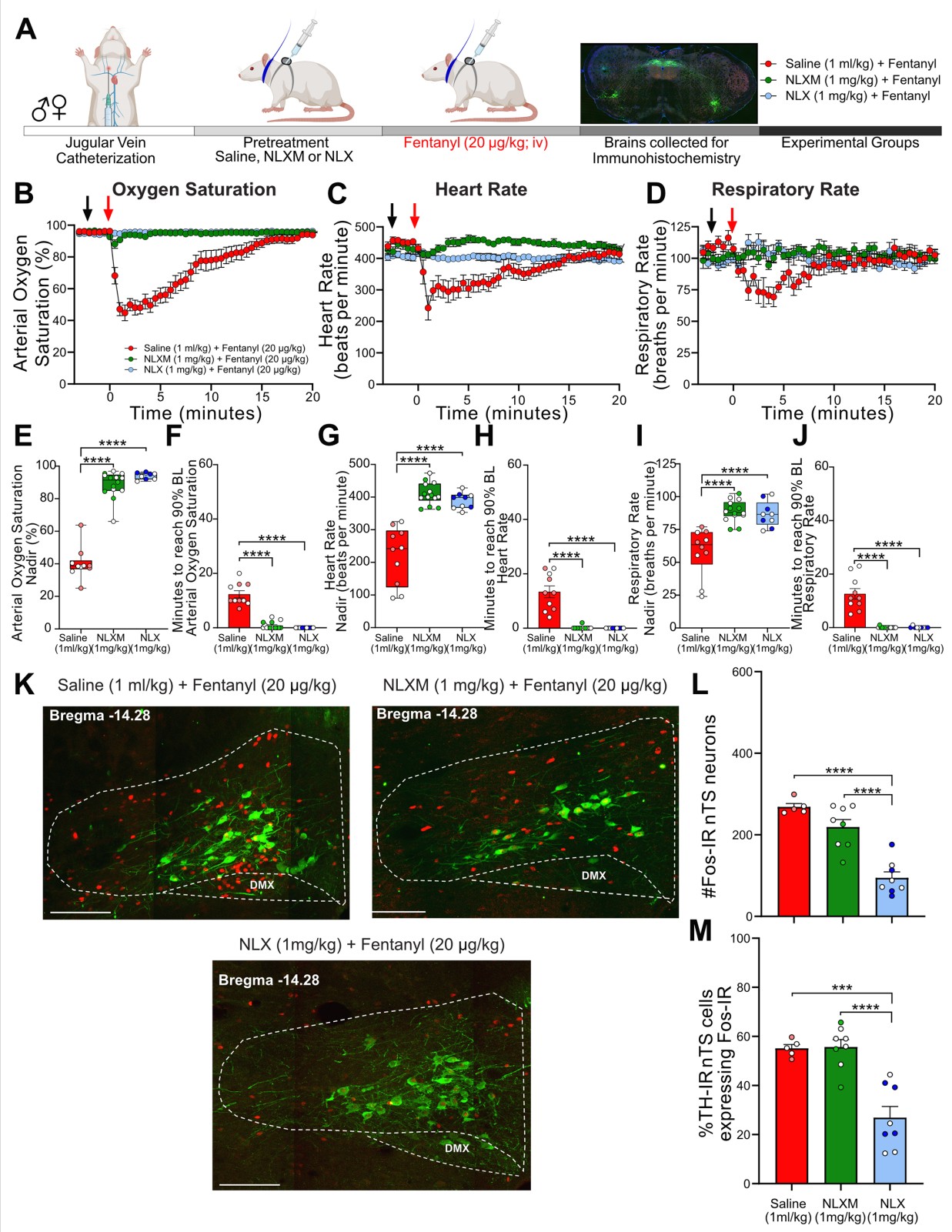

**Figure 3.** NLXM attenuates respiratory depression without altering nucleus of the solitary tract (nTS) Fos-IR induced by 20 µg/kg fentanyl. (**A**) Schematic showing the timeline for opioid-induced respiratory depression (OIRD) collar experiments. Male and female SD rats underwent jugular catheterization surgery. The following week, conscious rats in their home cages received intravenous administration (pretreatment) of either saline (1 ml/kg; $n$ = 10), NLX (1 mg/kg; $n$ = 9), or NLXM (1 mg/kg; $n$ = 15). Two minutes later, all rats received intravenous fentanyl (20 µg/kg). Cardiorespiratory parameters were

*Figure 3 continued on next page*

*Figure 3 continued*

measured up to 60 min after fentanyl. At the conclusion of the experiment, brains were collected from a subset of animals in each group and processed for Fos- and TH-immunoreactivity. Time course data showing oxygen saturation (**B**), heart rate (**C**), and respiratory rate (**D**) data measurements collected in all groups before and after saline, NLX, or NLXM pretreatment and after fentanyl. Changes in oxygen saturation were assessed as nadir (lowest value after fentanyl; **E, G, I**) and time to recover to 90% of baseline (**F, H, J**). In all graphs, males are represented by filled symbols, and females by open symbols. Mean nadir values were assessed in saline-, NLX-, and NLXM-pretreated rats (oxygen saturation: one-way ANOVA, $F_{(2,30)}$ = 152.0, p < 0.0001; Tukey's post hoc test, ****p < 0.0001; heart rate: one-way ANOVA, $F_{(2,30)}$ = 38.58, p < 0.0001; Tukey's post hoc test, ****p < 0.0001; respiratory rate: one-way ANOVA, $F_{(2,30)}$ = 20.16, p < 0.0001; Tukey's post hoc test, ****p < 0.0001). Time to reach 90% baseline in saline-, NLX-, and NLXM-pretreated rats. Mean time to reach 90% baseline was assessed for all parameters. Oxygen saturation: one-way ANOVA, $F_{(2,30)}$ = 81.9, p < 0.0001; Tukey's post hoc test ****p < 0.0001; heart rate: one-way ANOVA, $F_{(2,30)}$ = 44.11, p < 0.0001; Tukey's post hoc test ****p < 0.0001; respiratory rate: one-way ANOVA, $F_{(2,30)}$ = 46.74, p < 0.0001; Tukey's post hoc test ****p < 0.0001. (**K**) Merged photomicrographs of a coronal brainstem section showing representative Fos- and TH-immunoreactivity in the nTS of rats that received fentanyl with or without saline or NLX/NLXM pretreatment. Scale bar = 200 µm. Fos-IR and the percentage of TH-IR neurons expressing Fos-IR was evaluated in 3 nTS sections per rat. (**L**) Mean data show no significant difference in the number of Fos-IR cells between saline- and NLXM-pretreated rats. NLX-pretreated rats displayed the lowest degree of Fos-IR in the nTS. Number of Fos-IR cells: one-way ANOVA, $F_{(2,18)}$ = 30.52, p < 0.0001; Tukey's post hoc ****p < 0.0001. (**M**) Mean data of the percentage of TH-IR nTS cells expressing Fos-IR. The percentage of TH-IR cells expressing Fos-IR was similar between saline- and NLXM-pretreated rats. NLX-pretreated rats displayed the lowest percentage of Fos-IR in TH-IR nTS cells. Fos+TH/TH: one-way ANOVA, $F_{(2,18)}$ = 21.33, p < 0.0001; Tukey's post hoc ****p < 0.0001; ***p < 0.001.

The online version of this article includes the following source data and figure supplement(s) for figure 3:

**Source data 1.** Excel spreadsheet of statistical source data for *Figure 3*.

**Figure supplement 1.** NLXM attenuates cardiorespiratory depression induced by a higher dose of fentanyl.

**Figure supplement 1—source data 1.** Excel spreadsheet of statistical source data for *Figure 3—figure supplement 1*.

## Peripheral opioid receptor antagonism reverses fentanyl-induced respiratory depression

The above data demonstrate that peripheral MOR antagonism prevents fentanyl-induced respiratory depression without affecting nTS neuronal activation as measured by Fos-IR. Given that reversal of OIRD has more translational relevance, we next evaluated the extent to which NLXM reverses cardiorespiratory depression after it has been induced by 20 µg/kg (*Figure 4A*) or 50 µg/kg fentanyl (*Figure 4—figure supplement 1A*). Both 20 (*Figure 4B–D*) and 50 µg/kg fentanyl (*Figure 4—figure supplement 1B–D*) induced rapid decreases in oxygen saturation, heart rate, and respiratory rate, similar to what is observed in *Figure 1*. The fentanyl-induced nadir in all physiological parameters prior to intravenous administration of saline, NLX, or NLXM was similar between groups. In rats that received 20 µg/kg fentanyl, both intravenous NLX (1 mg/kg) and NLXM (1 mg/kg) restored cardiorespiratory values to baseline significantly faster than saline-pretreated animals (*Figure 4F, H, J*). This reversal of cardiorespiratory parameters was similar between NLX- and NLXM-treated animals. In rats that received 50 µg/kg fentanyl, NLX rapidly restored all physiological parameters to their baseline values (*Figure 4—figure supplement 1F, H, J*). While 1 mg/kg NLXM did not sufficiently reverse the OIRD induced by 50 µg/kg fentanyl (*Figure 4—figure supplement 1F*), a higher dose of 5 mg/kg NLXM fully reversed the fentanyl-induced bradycardia and hypoventilation. The time to restore parameters to baseline was not significantly different than NLX. We did not observe any sex differences in the nadir or recovery time of any parameter (*Supplementary file 1C, D*). Thus, blocking peripheral MORs sufficiently reverses OIRD at both moderate and high doses of fentanyl.

Because acute hypoxia activates nTS neurons (*Teppema et al., 1997*), we investigated the impact of peripheral and central MOR antagonism on fentanyl-induced nTS neuronal activation (*Figure 4K*). At both doses of fentanyl administered, a similar degree of nTS neuronal activation was observed in all groups regardless of MOR antagonist administered after fentanyl (*Figure 4L, M*; *Figure 4—figure supplement 1L, M*). Together, these data demonstrate that selective antagonism of peripheral opioid receptors sufficiently reverses cardiorespiratory depression induced by both moderate and high doses of fentanyl. Moreover, these data suggest that even a brief period of fentanyl-induced cardiorespiratory depression is sufficient to induce robust activation of nTS neurons, and this subsequent MOR antagonism has no additional effect to increase nTS activation.

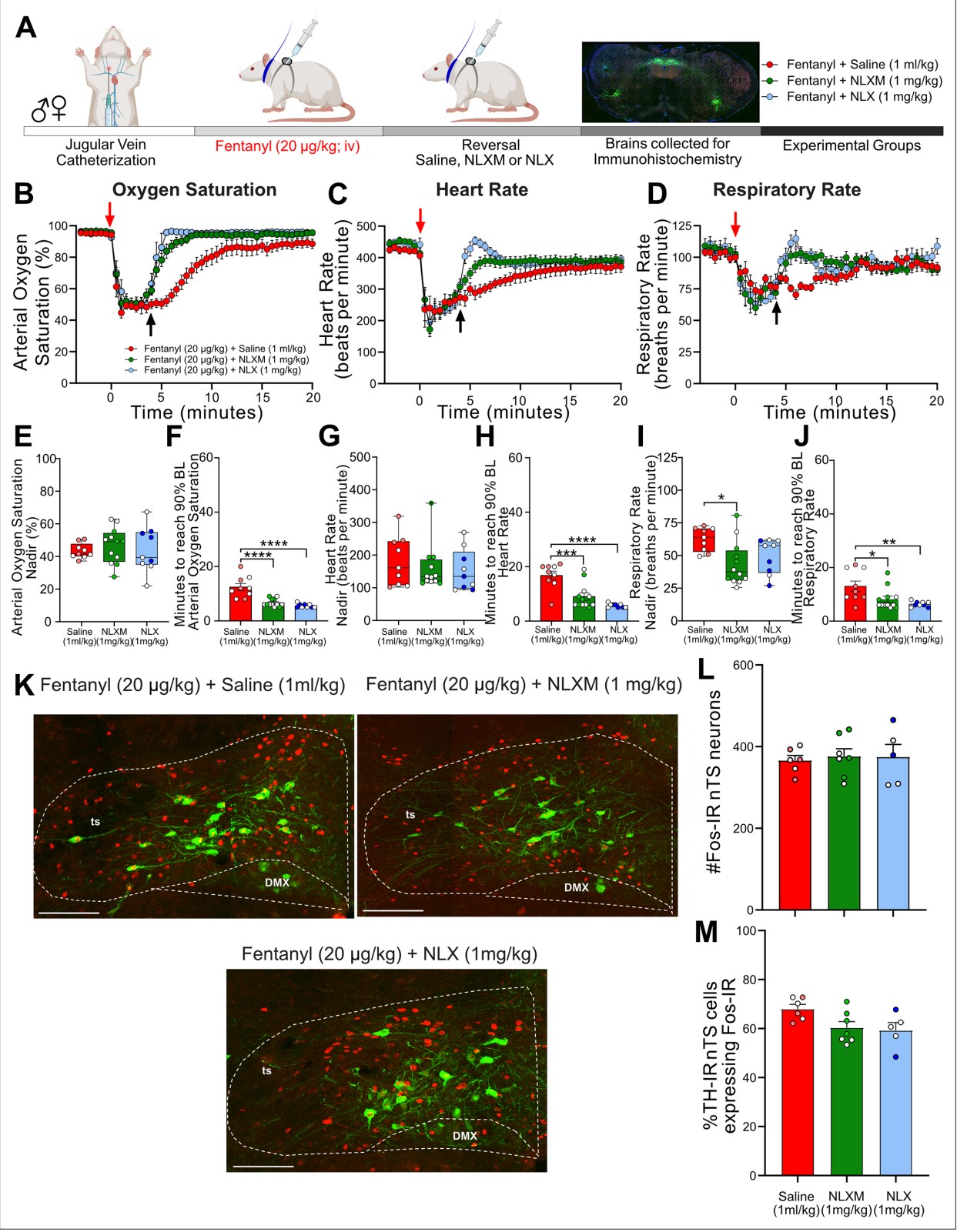

**Figure 4.** NLXM-mediated reversal of 20 µg/kg fentanyl-induced respiratory depression is comparable to NLX. (**A**) Schematic showing the timeline for opioid-induced respiratory depression (OIRD) reversal experiments. Male and female SD rats underwent jugular catheterization surgery. The following week, conscious rats in their home cages received intravenous fentanyl (20 µg/kg). Four minutes later, rats received intravenous administration (reversal) of either saline (1 ml/kg; *n* = 9), NLX (1 mg/kg; *n* = 9), or NLXM (1 mg/kg; *n* = 12). Cardiorespiratory parameters were measured up to 60 min

*Figure 4 continued on next page*

*Figure 4 continued*

after fentanyl. At the conclusion of the experiment, brains were collected from a subset of animals in each group and processed for Fos- and TH-immunoreactivity. Time course data showing oxygen saturation (**B**), heart rate (**C**), and respiratory rate (**D**) data measurements collected in all groups before and after fentanyl and saline/antagonist administration. Changes in oxygen saturation were assessed as nadir (lowest value; **E, G, I**) and time to recover to 90% of baseline (pre-saline/mu opioid receptor [MOR] antagonist; **F, H, J**). In all graphs, males are represented by filled symbols, and females by open symbols. Mean nadir values were assessed in saline-, NLX-, and NLXM-pretreated rats (oxygen saturation: one-way ANOVA, $F_{(2,27)}$ = 0.2916, p = 0.7494; heart rate: one-way ANOVA, $F_{(2,27)}$ = 0.4216, p = 0.6602; respiratory rate: one-way ANOVA, $F_{(2,27)}$ = 4.679, p = 0.018; Tukey's post hoc test, *p < 0.05. Mean time to reach 90% baseline was assessed for all parameters: oxygen saturation: one-way ANOVA, $F_{(2,27)}$ = 22.58, p < 0.0001; Tukey's post hoc test, ****p < 0.0001; heart rate: one-way ANOVA, $F_{(2,27)}$ = 19.39, p < 0.0001; Tukey's post hoc test, ****p < 0.0001, ***p < 0.001; respiratory rate: one-way ANOVA, $F_{(2,27)}$ = 6.322, p = 0.0001; Tukey's post hoc test, **p < 0.01, *p < 0.05). (**K**) Merged photomicrographs of coronal brainstem sections displaying representative Fos- and TH-immunoreactivity in the nucleus of the solitary tract (nTS) of rats that received fentanyl followed by saline, NLX, or NLXM. Scale bar = 200 μm. (**L**) Mean data of the number of Fos-IR cells evaluated in 3 nTS sections per rat. Number of Fos-IR cells: one-way ANOVA, $F_{(2,15)}$ = 0.07269, p = 0.9302. (**M**) Mean data of the percentage of TH-IR cells expressing Fos-IR: one-way ANOVA, $F_{(2,15)}$ = 3.736, p = 0.0482. There were no significant differences in the number of Fos-IR cells or the percentage of TH-IR cells expressing Fos-IR between groups.

The online version of this article includes the following source data and figure supplement(s) for figure 4:

**Source data 1.** Excel spreadsheet of statistical source data for *Figure 4*.

**Figure supplement 1.** NLXM reverses cardiorespiratory depression induced by a higher dose of fentanyl.

**Figure supplement 1—source data 1.** Excel spreadsheet of statistical source data for *Figure 4—figure supplement 1*.

## Reversal of fentanyl-induced respiratory depression with NLXM is not aversive

NLX rapidly reverses OIRD, but produces unwanted side effects including the immediate onset of severe withdrawal (*Lewanowitsch and Irvine, 2002*; *Lai et al., 2021*). The intense symptoms induced during NLX-precipitated withdrawal present additional challenges in managing OUD in patients. Given that peripheral MOR antagonism sufficiently reverses fentanyl-induced cardiorespiratory depression to a degree that is comparable to NLX, we used a conditioned place aversion (CPA) approach to assess whether selective peripheral antagonism of MOR signaling produces aversive behaviors. Rats underwent a three-session CPA challenge that consisted of pre-conditioning test, conditioning test, and post-conditioning test (*Figure 5A*). Horizontal locomotion in both compartments of the CPA boxes was assessed during both pre- and post-conditioning tests (*Figure 5B*). During the pre-conditioning test, rats exhibited no natural preference or aversion for either compartment. On the conditioning day (Day 2), a divider was placed in the middle of the CPA box. Rats received intravenous fentanyl (50 μg/kg) in their home cage. Four minutes later, rats were placed into the paired compartment of the CPA box and immediately given intravenous saline (1 ml/kg), NLX (1 mg/kg), or NLXM (1 or 5 mg/kg). During the post-conditioning test, the divider was removed, and rats were allowed to explore both compartments of the CPA box. *Figure 5C* shows the time spent on the conditioned side of the CPA box for all groups. During the post-conditioning test, NLX-treated animals spent significantly less time in the NLX-paired compartment, suggesting that the reversal of OIRD on the conditioning day was aversive. This CPA behavior was not observed in rats that received NLXM at either dose (*Figure 5D*). Importantly, a single conditioning session with fentanyl administered in the home cage followed by saline in the compartment instead of an MOR antagonist did not produce a preference or aversion to the saline-paired compartment. To rule out any potential effects of endogenous opioids influencing the behavioral responses to opioid receptor antagonism, we performed an additional test in which rats were given saline in their home cage, followed by NLX or NLXM in one of the CPA compartments (*Figure 5E, F*). Neither antagonist produced a preference nor aversion to the MOR antagonist-paired compartment (*Figure 5G*). There was no difference in CPA scores between these groups (*Figure 5H*). These findings are consistent with our data demonstrating that NLX and NLXM alone do not alter physiological parameters from baseline (*Figure 2—figure supplement 1*).

## Fentanyl induces a biphasic activity profile in nTS neurons

As shown in *Figure 1*, fentanyl-induced robust Fos-IR expression in the nTS. While Fos-IR quantification provides insight into the anatomical activation of brainstem subregions in response to fentanyl administration and OIRD reversal, this approach does not provide any temporal resolution of the activity of these cells. Therefore, we used wireless in vivo fiber photometry to assess dynamic changes in nTS

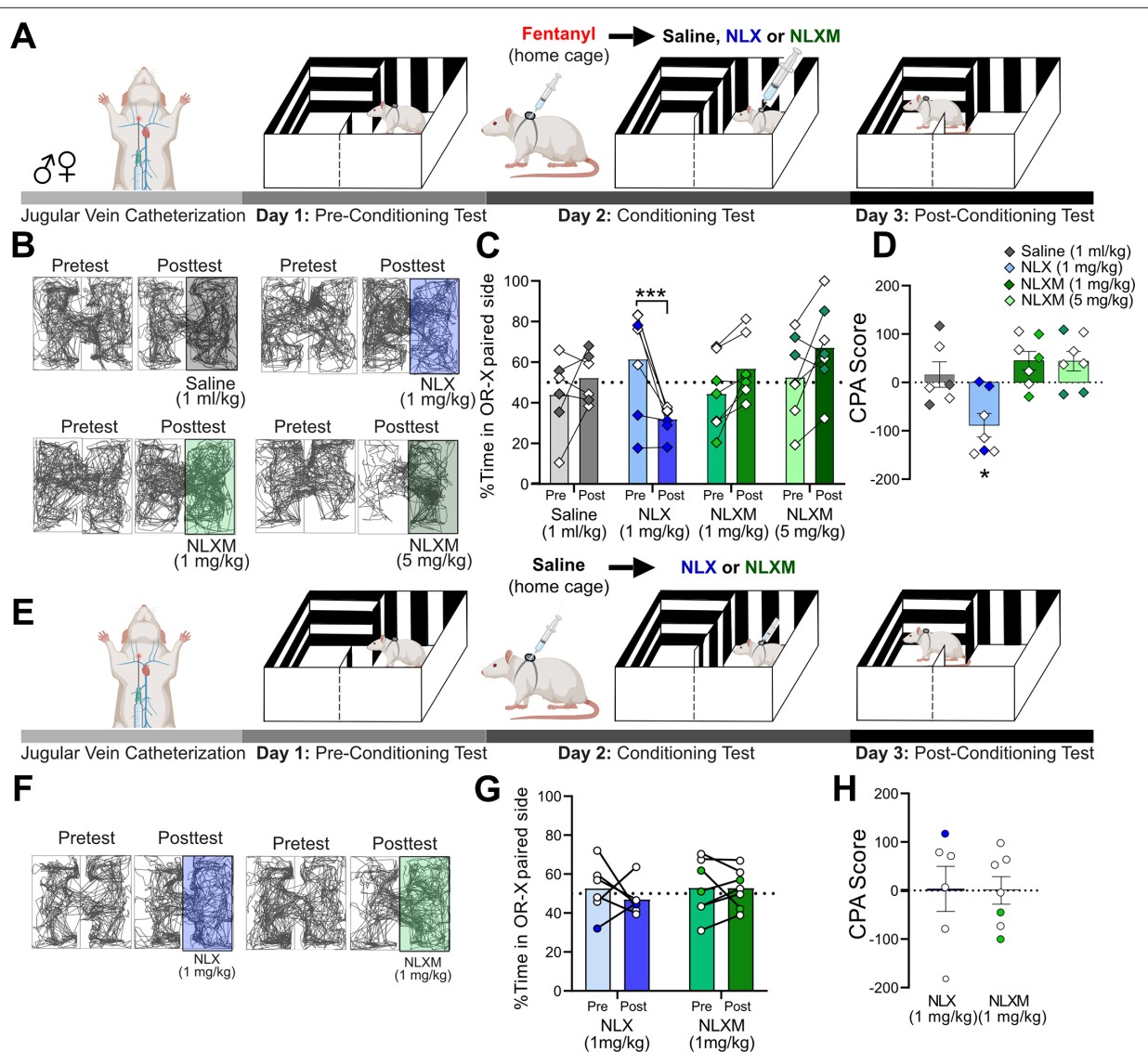

**Figure 5.** NLXM-mediated reversal of opioid-induced respiratory depression (OIRD) is not aversive. (**A**) Schematic of the conditioned place aversion (CPA) experimental design for rats that received fentanyl followed by saline or an opioid receptor antagonist. Catheterized rats underwent a pre-conditioning test (Day 1). On the conditioning day (Day 2), a divider was placed in the middle of the CPA box. Rats received intravenous fentanyl (50 μg/kg) in their home cages. Four minutes later, rats were transferred to one side of the CPA box and immediately received intravenous saline (1 ml/kg, *n* = 6), NLX (1 mg/kg, *n* = 7), or NLXM (1 mg/kg, *n* = 7; 5 mg/kg, *n* = 7). During the post-conditioning test (Day 3), the divider was removed, and rats were placed in the box and allowed to explore both compartments. (**B**) Track plot data showing movement in both sides of the CPA box during the Pretest (Day 1) and Posttest (Day 3). For each condition, the Pre- and Posttest data are from the same animal. In these representative examples, the saline- or antagonist-paired side is shown on the right side of the box. However, the paired side was balanced between left and right sides of the box for all groups. (**C**) Mean data with individual plots showing changes in the time spent in the saline- and antagonist-paired side. In all graphs, males are represented by filled symbols, and females by open symbols. There was a significant decrease in the time spent in the NLX-paired side: two-way repeated measures ANOVA; drug × test interaction: $F_{(3,23)}$ = 9.994, p = 0.0002; Sidak's post hoc, \*\*\*p = 0.0007. (**D**) The CPA score calculated as time spent in the saline or antagonist paired compartment during posttest minus the time spent in the same compartment during the pretest. For CPA scores, one-sample *t*-test from hypothetical value (no aversion). There was a significant decrease in the CPA score for 1 mg/kg NLX-treated rats *t* = 3.642 (df = 6, \*p < 0.05). 1 mg/kg NLXM: *t* = 2.340, df = 6, p = 0.0578; 5 mg/kg NLXM: *t* = 2.172, df = 6, p = 0.0729. Saline, *t* = 0.6113 (df = 5, p = 0.5677). (**E**) Schematic of the CPA experimental design for saline-pretreated animals. Catheterized rats underwent a pre-conditioning test (Day 1). On the conditioning day (Day 2), a divider was placed in the middle of the CPA box. Rats received intravenous saline (1 ml/kg) in their home cages. Four minutes later, rats were transferred to one side of the CPA box and immediately received intravenous NLX (1 mg/kg, *n* = 6) or NLXM (1 mg/kg, *n* = 7). (**F**) Track plot data showing movement in both sides of the CPA box during the Pretest (Day 1) and Posttest (Day 3). The paired side was balanced between left and right sides of the box for all groups. (**G**) Mean data with individual plots showing changes in the time spent on the NLX- or NLXM-paired side. There was no difference in time spent in the side paired with NLX or NLXM (two-way repeated measures ANOVA, antagonist × time, $F_{(1,11)}$

*Figure 5 continued on next page*

*Figure 5 continued*

= 0.4638, p = 5.099). (**H**) CPA scores for NLX- and NLXM-treated animals. There was no significant difference in the CPA scores in rats that received NLX (one-sample *t*-test, *t* = 0.07102, df = 5, p = 0.9461) or NLXM (one-sample *t*-test, *t* = 0.005032, df = 6, p = 0.9961).

The online version of this article includes the following source data for figure 5:

**Source data 1.** Excel spreadsheet of statistical source data for *Figure 5*.

calcium transient activity after intravenous fentanyl administration in anesthetized rats (*Figure 6A*). Catheterized rats underwent bilateral injections of an AAV expressing a calcium sensor (pGP-AAV9-syn-jGCaMP8m-WPRE) into the nTS. On the day of the experiment, an optic fiber was lowered on the surface of the brainstem above the nTS and calcium transient activity was recorded (*Figure 6B*). GCaMP viral expression was observed throughout the caudal-rostral extent of the nTS in all rats (*Figure 6C*). Fentanyl (20 µg/kg) infusion induced a biphasic activity profile in nTS neurons, characterized by an initial robust increase (Early Response) followed by a decrease below baseline (Late Response, *Figure 6D*). Pretreatment with NLXM (5 mg/kg) attenuated both phases of the fentanyl-induced changes in neuronal activity. As shown in *Figure 6E*, the area under the curve (AUC) was significantly elevated during the Early and Late Responses after fentanyl. In contrast, no significant fentanyl-induced change in AUC was observed in NLXM-pretreated rats at either time point, indicating that changes in fentanyl-induced nTS neuronal activity are strongly mediated by peripheral MORs. We next examined the extent to which MOR antagonists restore the fentanyl-induced reductions in activity. After MOR antagonist administration, fentanyl-induced suppression of nTS activity returned to baseline during the early recovery in rats that received NLX, but not NLXM. However, both NLX and NLXM enhanced nTS activity above baseline during the late recovery period (*Figure 6F, G*). During the Late Recovery period, nTS activity was significantly enhanced above baseline in both NLX- and NLXM-treated animals. Together, these data demonstrate that peripheral MOR antagonism is sufficient to reverse the fentanyl-induced decrease in nTS neuronal activity.

## Discussion

OIRD is a complex problem that continues to be at the forefront of the opioid epidemic (*Ramirez et al., 2021*; *Bateman et al., 2023*). The increased potency and faster onset of synthetic opioids like fentanyl has been attributed to the rising wave of opioid-related deaths in the US (*Comer and Cahill, 2019*; *Fairbairn et al., 2017*). Numerous studies have evaluated opioid-induced dysfunction within key regulatory sites in the central nervous system that contribute to the severity of OIRD (*Ramirez et al., 2021*; *Bachmutsky et al., 2020*; *Bateman and Levitt, 2023*; *Levitt et al., 2015*; *Maletz et al., 2022*). Peripheral MORs have also been examined for their roles in impacting OIRD (*Lewanowitsch and Irvine, 2002*; *Henderson et al., 2013*). However, the specific contributions of central and peripheral MORs to the onset and duration of fentanyl-induced cardiorespiratory depression, along with the role of the nTS in these effects, are not completely understood. In this study, we show that blocking peripheral MORs with NLXM sufficiently prevents and reverses fentanyl-induced cardiorespiratory depression and systemic hypoxia. Furthermore, we demonstrate that peripheral opioid receptors mediate fentanyl-induced activity of nTS neurons. These data suggest that the nTS may be an area impacted by fentanyl via peripheral MOR activation. Lastly, we demonstrate that in addition to reversing fentanyl-induced cardiorespiratory depression, NLXM does not induce aversive behaviors, as compared to NLX. Together, these findings provide evidence that peripheral MOR antagonism may be a potential novel strategy to reverse OIRD without inducing withdrawal, anxiety, and aversion, all of which can contribute to further relapse in drug-seeking behaviors (*Comer and Cahill, 2019*; *Lai et al., 2021*).

While numerous studies have focused on evaluating the effects of opioids acting at MORs present within brainstem and pontine networks, less attention has been given to identifying peripheral MOR mechanisms contributing to OIRD. NLX is the most common reversal agent used to treat OIRD (*Lai et al., 2021*; *Bateman et al., 2023*). In addition to reversing opioid-induced analgesia, NLX precipitates withdrawal symptoms and aversion in humans and rodents (*Lewanowitsch and Irvine, 2002*; *Lai et al., 2021*; *Neale and Strang, 2015*; *Dahan et al., 2018*). In this study, we demonstrate that NLX-mediated reversal of OIRD caused a CPA, as shown by a significant decrease in time spent in the NLX-associated compartment. This is likely the result of the induction of withdrawal symptoms that

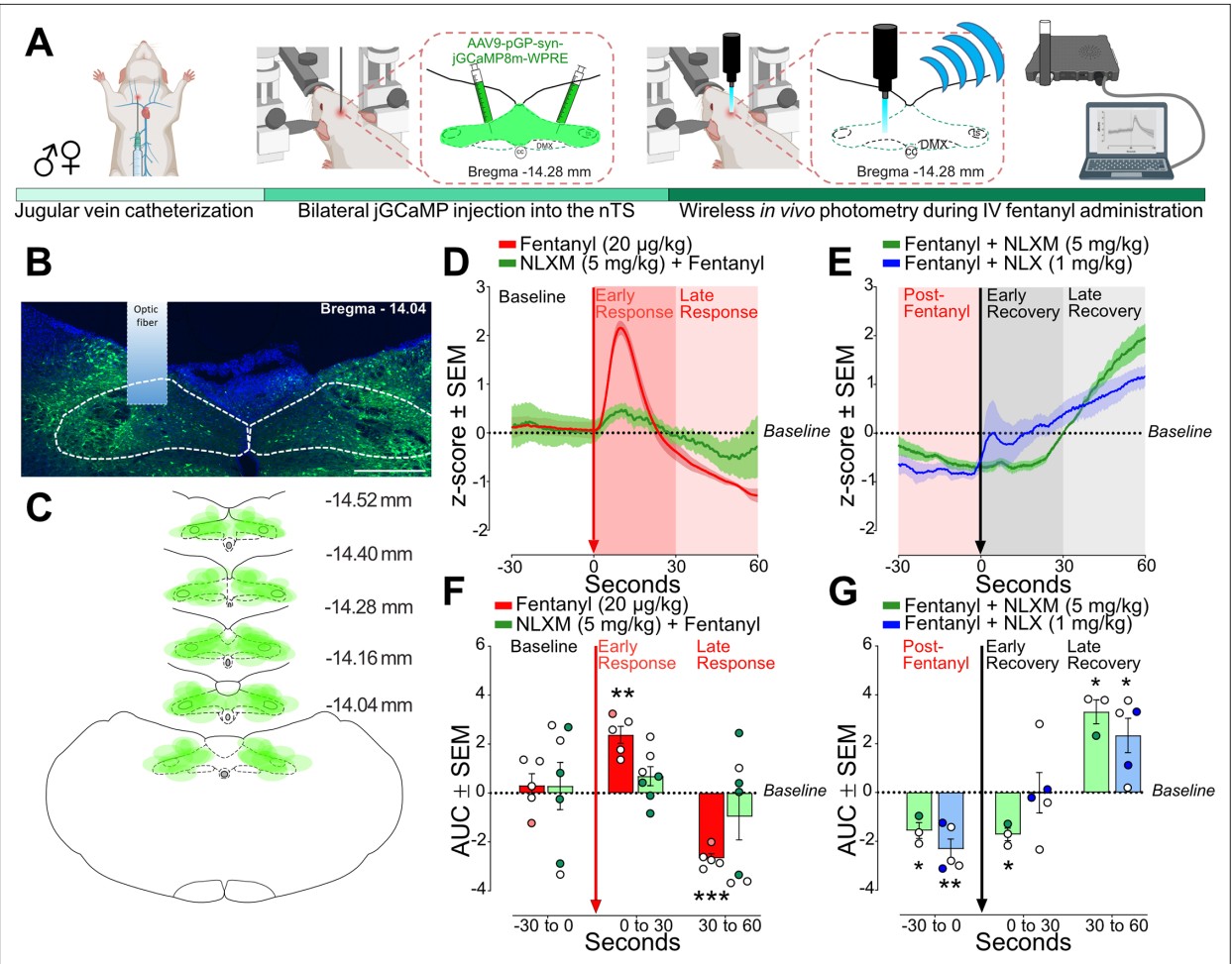

**Figure 6.** Fentanyl-induced nucleus of the solitary tract (nTS) activity is mediated by peripheral mu opioid receptors (MORs). (**A**) Experimental timeline. Male and female rats underwent jugular catheterization surgery, followed by bilateral nanoinjection of jGCaMP8m into the nTS and then allowed 3–5 weeks for viral expression. On the test day, rats were anesthetized and placed in a stereotaxic apparatus. The nTS was exposed and a fiber was lowered into the nTS. (**B**) Representative example of GCaMP expression and fiber placement in the nTS. Scale bar = 200 µm. (**C**) Spread of overlay of individual animal viral expression across the caudal-rostral extent of the nTS. (**D**) nTS cell response (mean z-score) aligned to 20 µg/kg fentanyl administration (red arrow) at Baseline (−30 to 0 s), Early Response (0–30 s), and Late Response (30–60 s) in rats that received 20 µg/kg fentanyl only (red trace) or 5 mg/kg NLXM followed by 20 µg/kg fentanyl (green trace). In all graphs, males are represented by filled symbols, and females by open symbols. (**E**) Area under the curve (AUC) was calculated during baseline (−30 to 0 s), Early Response (0–30 s), or Late Response (30–60 s). For AUC graphs, one-sample $t$-test from hypothetical zero value. In rats that received fentanyl only, there was a significant increase in the AUC during the Early Phase ($t$ = 6.747, df = 4, **p = 0.0025) followed by a significant decrease in the Late Phase ($t$ = 14.35, df = 4, ***p = 0.0001). In rats that received 5 mg/kg NLXM prior to 20 µg/kg fentanyl, no differences in AUC were observed at any time point (Baseline; $t$ = 0.2916, df = 6, p = 0.7804; Early Phase; $t$ = 1.760, df = 6, p = 0.1288; 30–60 s; $t$ = 1.002, df = 6, p = 0.3548). (**F**) nTS cell response (mean z-score) aligned to MOR antagonists (black arrow). Rats received iv 20 µg/kg fentanyl (Post-Fentanyl phase, −30 to 0 s) prior to MOR antagonist infusion of NLX (1 mg/kg, blue trace) or NLXM (5 mg/kg, green trace). Data during Early Recovery (0–30 s) and Late Recovery (30–60 s) are shown. (**G**) Mean data showing AUC analyses for both groups. For all graphs, one-sample $t$-test from hypothetical zero value. NLXM-treated rats had a significant decrease in activity from Post-Fentanyl baseline prior to NLXM ($t$ = 4.799, df = 2, *p = 0.0408) and during the Early Recovery phase ($t$ = 6.683, df = 2, p = 0.0217), and exhibited enhanced activity above baseline during the Late Recovery phase ($t$ = 6.716, df = 2, *p = 0.0215). NLX-treated rats displayed a similar decrease below baseline prior to NLX. No difference from baseline was observed during the Early Recovery phase ($t$ = 0.01238, df = 4, p = 0.9907), and a significant increase above baseline was observed during the Late Recovery phase ($t$ = 3.318, df = 4, *p = 0.0294).

The online version of this article includes the following source data for figure 6:

**Source data 1.** Excel spreadsheet of statistical source data for **Figure 6**.

occur following antagonism of MOR in central reward pathways (*Lewanowitsch and Irvine, 2002*; *Lai et al., 2021*; *Katovich et al., 1986*). In contrast, this aversive response was not observed in rats that received NLXM. These data suggest that alleviating fentanyl-induced respiratory distress without antagonizing central MOR may mitigate the aversive state produced by NLX.

Conditioned place preference and aversion paradigms have been utilized to evaluate rewarding and aversive properties of opioids and antagonists (*Finlay et al., 1988*; *Gaulden et al., 2021*; *Knauss et al., 2023*). Overall, we observed relatively minimal sex differences in basal and fentanyl-induced changes in cardiorespiratory depression, which is consistent with a previous report showing that sex differences due to fentanyl are relatively minimal compared to other opioids such as heroin (*Marchette et al., 2023*). Previous reports have shown sexually dimorphic differences in fentanyl-induced conditioned place preference, with higher doses of fentanyl affecting females only (*Knauss et al., 2023*). In present study, we administered fentanyl to male and female rats in the home cage followed by saline in the CPA box and observed neither a preference nor aversion in either sex. This suggests that the results obtained during OIRD-mediated reversal via NLX or NLXM are driven by effects within the CPA box and are not influenced by the rewarding properties of fentanyl itself. While previous reports have shown that NLX can be detected centrally after a subcutaneous injection of NLXM (*Perekopskiy et al., 2020*), we observed no detectable amount of NLX or NLXM in brain tissue of rats that received intravenous NLXM at either dose or time point. Our data demonstrate robust fentanyl-induced neuronal activation in the nTS that appears to be mediated via peripheral MORs. Given the proximity of the nTS to the circumventricular organ area postrema (*Price et al., 2008*), this raises the possibility that fentanyl can access the central nervous system via the area postrema to influence neuronal activation. However, our data showing the lack of a CPA following NLXM-mediated reversal of OIRD support our biodistribution data showing no NLXM detection in brain samples that the observed effects primarily involve fentanyl actions at MOR located in peripheral sites. Similar to NLX, peripheral MOR antagonism with NLXM has been shown to partially reverse opioid-induced analgesia (*Lewanowitsch and Irvine, 2002*; *Spahn et al., 2017*) which may pose limitations for pain management. While we report no changes in cardiorespiratory parameters from baseline following intravenous administration of NLX or NLXM in drug-naive animals, previous work has shown that high doses of NLX elicit behavioral and physiological changes in humans, including cognitive impairment, nausea, tachycardia, and hypertension, even at low and moderate doses (*Yugar et al., 2023*; *Cohen et al., 1981*; *Cohen et al., 1982*). Future investigations are needed to evaluate mechanisms underlying the therapeutic potential of NLXM as an intervention for managing OIRD in patients with OUDs.

The nTS is a region with a high degree of MOR expression, and appears to be primarily located on afferent fibers arising from the vagus nerve (*Zhuang et al., 2017*; *Aicher et al., 2000*; *Nomura et al., 1996*). While MOR has been shown to be present in nTS somas and dendrites (*Aicher et al., 2000*; *Maletz et al., 2022*), these MOR-expressing cells do not express Fos following exposure to opioids or hypoxia (*Maletz et al., 2022*), which suggests a mechanism of presynaptic action of MOR signaling in the nTS. Likewise, the nodose ganglion, which contains the cell bodies of nTS-projecting vagal sensory afferents, has high expression of MOR (*Li et al., 1996*), and intra-nodose injection of fentanyl induces hypoventilation, although to a much lesser degree compared to systemic fentanyl (*Zhang et al., 2012*).

In the present study, we used fiber photometry to characterize nTS neuronal responses to show that fentanyl induces an initial excitatory response in nTS cells. However, this response was short-lasting compared to the subsequent decrease in activity below baseline, suggesting that the primary effect of MOR signaling produce generalized inhibition of nTS neurons. This prolonged suppression of nTS neuronal activity may contribute to the overall duration of OIRD. Furthermore, our photometry data demonstrate that while the NLXM-mediated reversal of fentanyl-induced inhibition of nTS neurons was slower compared to NLX, activity was eventually restored. A similar finding was observed in OIRD experiments, as NLXM induced a rate of recovery in all parameters that was comparable to NLX. Taken together, this suggests that peripheral MOR antagonism may sufficiently override simultaneous central effects of fentanyl acting at MOR present in cardiorespiratory nuclei, including the nTS.

It is important to note that while all our experiments assessing fentanyl-induced cardiorespiratory depression were conducted in conscious rats breathing room air, photometry experiments were performed under isoflurane anesthesia. Due to the location of the nTS, chronic fiber implantation and subsequent in vivo photometry recordings in conscious, unrestrained animals are not possible.

In the nTS, Isoflurane has been reported to suppress visceral afferent glutamatergic transmission and enhance GABAergic transmission via postsynaptic mechanisms (*Peters et al., 2008*), as well as induce cFos expression in the nTS (*Hu et al., 2024*). Precautions were taken to use the lowest dose of isoflurane needed to maintain sufficient anesthesia. However, we cannot rule out the possible influence of isoflurane anesthesia to influence the observed biphasic responses evoked by fentanyl. In addition, sex differences in physiological and pharmacological responses to anesthesia have been previously described (*Mawhinney et al., 2013*). Due to the high attrition rate for these experiments, which require multiple survival surgeries and proper fiber placement into the nTS, we were unable to fully evaluate potential sex differences and cannot rule out the possibility of a sex effect of fentanyl-induced neural activity or in responses to anesthesia. Nevertheless, our data support a mechanism by which fentanyl induces OIRD via disruption of visceral afferent signaling to the nTS, thereby leading to prolonged OIRD due to suppression of physiological reflex responses that would be normally engaged in response to systemic hypoxia.

The nTS plays an essential role in the regulation of basal- and reflex-evoked cardiorespiratory function (*Andresen and Kunze, 1994*). Activation of MOR in the nTS has been shown to blunt hypoxic ventilatory responses following MOR agonism in the nTS (*Zhuang et al., 2017*). Our data are consistent with previous work demonstrating opioid-induced activation of nTS neurons (*Maletz et al., 2022*). The degree of fentanyl-induced Fos-IR in the nTS resembles what is observed after acute exposure to hypoxic air (*Montandon et al., 2011*; *Cohen et al., 1981*). Interestingly, we also observed increased Fos in the nTS at a dose of fentanyl that was below the threshold to induce respiratory depression. It is unclear whether the Fos-IR observed after low or high doses of fentanyl is, in part, the result of increased glutamatergic transmission stemming from enhanced chemoreceptor afferent signaling to the nTS or via MOR signaling within the nTS itself.

Glutamate is the primary neurotransmitter released from sensory afferent fibers into the nTS (*Andresen and Kunze, 1994*; *Talman et al., 1980*). While intra-nTS application of opioid agonists have been shown to reduce glutamate release and calcium channels via presynaptic mechanisms (*Cui et al., 2012*; *Rhim and Miller, 1994*), opioids also have been shown to induce a similar biphasic action of $Ca^{2+}$-dependent spikes in the nodose ganglion (*Higashi et al., 1982*) that resembles what we observed in GCaMP-expressing nTS cells after intravenous fentanyl. NLXM strongly attenuated the fentanyl-induced excitatory wave observed in GCaMP-expressing nTS neurons, suggesting that MOR-expressing peripheral afferents mediate this transient activation of nTS neurons, even if the primary effect of fentanyl is to inhibit the nTS and suppress cardiorespiratory reflex function.

One possibility is that the increase in nTS activity may be mediated by opioid-induced disinhibition within the nTS. The nTS contains a large population of second-order GABAergic interneurons that receive direct inputs from visceral afferents (*Bailey et al., 2008*), and hypoxia activates these cells (*King et al., 2012*). nTS MOR agonists suppress the activity of these cells, primarily via presynaptic mechanisms (*Boxwell et al., 2013*), although postsynaptic mechanisms have been reported (*Glatzer and Smith, 2005*). Similarly, visceral afferents also activate catecholaminergic nTS neurons (*Appleyard et al., 2007*), which are critical for the generation of hypoxia-evoked cardiorespiratory responses (*King et al., 2015*; *Bathina et al., 2013*). Opioids inhibit visceral afferent transmission to this cell group, primarily via a presynaptic mechanism (*Cui et al., 2012*). In the present study, we observed robust Fos expression in these cells at all doses of fentanyl examined. These findings raise the possibility that nTS dysfunction, via MOR-expressing peripheral inputs that terminate within this region, represents the first central site of fentanyl-induced dysfunction, ultimately leading to downstream effects in other brain regions that contribute to diminished autonomic and cardiorespiratory responses that contribute to the duration of OIRD. Future studies are needed to fully evaluate the exact opioid-MOR effects in the nTS, including specific MOR afferent inputs and the nTS phenotypes, to determine their relative contributions to OIRD.

In summary, these studies highlight a significant involvement of peripheral MORs contributing to the rapid cardiorespiratory depression induced by intravenous fentanyl. We have provided extensive characterization into how fentanyl elicits cardiorespiratory depression and examined the relationship between peripheral MORs and nTS activity in mediating the duration of OIRD. Moreover, our findings suggest that selectively blocking peripheral opioid receptors could be a promising therapeutic strategy for managing OIRD. Notably, peripheral opioid receptor antagonists are already prescribed for patients undergoing chemotherapy to alleviate side effects such as opioid-induced constipation

(*Floettmann et al., 2017*; *Pergolizzi et al., 2020*). Importantly, our data demonstrate that peripheral MOR antagonism appears to reverse OIRD without triggering unwanted effects of withdrawal that occur following NLX administration, suggesting that peripheral MOR antagonists like NLXM could potentially be utilized in treatment strategies for managing patients with OUD.

# Materials and methods

**Key resources table**

| Reagent type (species) or resource | Designation | Source or reference | Identifiers | Additional information |
|---|---|---|---|---|
| Strain, strain background (Sprague Dawley, *Rattus norvegicus*) | Wild-type Sprague Dawley | Maintained In House | RRID:RGD_70508 | Male and female rats |
| Antibody | anti-c-Fos antibody (Rabbit polyclonal) primary antibody | Abcam | Cat# ab190289; RRID:AB_2737414 | IHC (1:2000) |
| Antibody | anti-TH (Mouse monoclonal) primary antibody | Abcam | Cat# MAB318; RRID:AB_2201528 | IHC (1:2000) |
| Antibody | Donkey anti-Rabbit Cy3 | Jackson ImmunoResearch Labs | Cat# 711-167-003; RRID:AB_2340606 | IHC (1:200) |
| Antibody | Donkey anti-Mouse AF488 | Jackson ImmunoResearch Labs | Cat# 115-545-150; RRID:AB_2340846 | IHC (1:200) |
| Other | pGP-AAV9-syn-jGCaMP8m-WPRE | Addgene | Cat # 162375-AAV9; RRID:Addgene_162375 | Adeno-asspciated virus to express jGCaMP8 in vivo. |
| Chemical Compound, drug | Naloxone hydrochloride dihydrate | Sigma | Cat#: N7758; CAS: 51481-60-8 | 1 mg/kg |
| Chemical compound, drug | Naloxone methiodide | Sigma | Cat#: N129 CAS: 93302-47-7 | 1 and 5 mg/kg |
| Software, algorithm | ImageJ V153.e | *Schneider et al., 2012* | RRID:SCR_003070 | https://imagej.net |
| Software, algorithm | Telefipho Software | Amuza | Cat# E59.100.00 | |
| Software, algorithm | MATLAB | Mathworks | RRID:SCR_001622 | |
| Software, algorithm | GraphPad Prism | GraphPad | RRID:SCR_002798 | |
| Software, algorithm | ANY-maze | Stoelting Europe | RRID:SCR_014289 | https://www.any-maze.com/ |
| Software, algorithm | Wireless Photometry Analysis Codes for MATLAB | This paper | | MATLAB code used to perform area under the curve analyses for fiber photometry experiments; https://doi.org/10.5281/zenodo.15042135 |

All surgical and experimental procedures were approved by Washington University Committee in accordance with the National Institutes of Health Guidelines for the Care and Use of Laboratory Animals and Animal Research: Reporting In Vivo Experiments (ARRIVE) guidelines. Rats were initially group-housed with two to three animals per cage on a 12/12-hr dark/light cycle (lights on at 7:00) and acclimated to the animal facility holding rooms for at least 7 days before any manipulation. Following catheterization surgery, rats were single-housed and remained on an identical 12/12-hr dark/light cycle. The temperature for the holding rooms of all animals ranged from 21 to 24°C while the humidity was between 30 and 70%. Rats received food and water ad libitum for the duration of all experiments. All experiments were performed during the light cycle. For all experiments, male and female Sprague Dawley rats (RRID:RGD_70508; 250–350 g) were used. In all datasets, male rats are represented by filled symbols and female rats are represented by open symbols.

## Jugular catheterization procedure

Animals were deeply anesthetized using isoflurane (3% induction, 2% maintenance). A small incision was made in the dorsal surface of the neck. Another small incision was made on the ventral surface

of the neck, and underlying tissue was carefully dissected to expose the jugular vein. An indwelling catheter was inserted into the jugular vein and sutured in place using non-absorbable silk suture. The catheter was then tunneled subcutaneously and exited the body through the first small hole in neck on the dorsal side. The ventral incision was closed using non-absorbable silk suture. The exposed catheter was connected to a backpack device (Instech) containing a port for drug administration. Animals were given Carprofen (2 mg/kg sc), Baytril (8 mg/kg sc), and bupivacaine (5 mg/kg, site of incision). In addition, rats were given Carprofen tablets (Bio Serv, MD150-2) for 2 days after surgery to assist in wound healing and analgesia. Rats were singly housed following surgery and were allowed to recover for 1 week prior to random assignment to an experimental group. Catheter patency was maintained with daily flushing of 0.3 ml sterile saline containing gentamicin (1.33 mg/ml, i.v.). Rats with a loss of catheter patency were excluded from the study.

## Respiratory depression experiments

Following recovery from IV catheterization surgery, conscious, freely moving rats in their home cages were fitted with a non-invasive pulse oximeter collar (Starr LifeSciences). Cardiorespiratory parameters (oxygen saturation, heart rate and respiratory rate) were collected (MouseOx v2.0, Starr LifeSciences) at baseline conditions in animals breathing room air, and after intravenous administration of fentanyl citrate (2, 20, or 50 µg/kg). A subset of animals received weight-adjusted intravenous administration of the nonselective opioid receptor antagonist naloxone hydrochloride (NLX; 1 mg/kg; Sigma, Cat#: N7758, CAS: 51481-60-8) or the peripherally restricted opioid receptor antagonist naloxone methiodide (NLXM; 1 or 5 mg/kg; Sigma, Cat#: N129, CAS: 93302-47-7) before or after fentanyl administration. Cardiorespiratory parameters were then measured for up to 60 min. For all OIRD experiments, the time course graphs consist of 30 s of data averaged into a single data point. The nadir was defined as the lowest point measured after fentanyl administration (for all graphs, red arrow indicates time of fentanyl administration, time = 0 min). In addition, the recovery rate was defined as the time to reach 90% baseline of pre-fentanyl values for each cardiorespiratory parameter. Two hours after fentanyl administration, rats were transcardially perfused with 0.01 M PBS followed by paraformaldehyde (4% in 0.01 M PBS). Brains were removed and stored overnight at 4°C in 4% paraformaldehyde for post-fixation, followed by at least 72-hr incubation in sucrose (30% in 0.01 M PBS) solution. Isopentane was used to flash-freeze brains. Brainstem sections containing the nTS were sliced in the coronal plane (40 µm) using a cryostat (Leica CM 1950).

## Biodistribution

A subset of previously catheterized drug-naive rats was deeply anesthetized and then received intravenous administration of NLX (1 mg/kg) or NLXM (1 or 5 mg/kg). While still under anesthesia, rats were quickly decapitated 2 or 10 min after intravenous administration of MOR antagonists. MOR antagonists were dosed in all rats at an experimentally relevant concentration with triplicate blood and brain samples collected at two time points. The early time point at 2 min evaluated conditions when the plasma concentration of the OR antagonist is high and reflects the identical time point of intravenous administration for OIRD pretreatment experiments. The second collection time point (10 min) was intended to evaluate steady-state conditions and represent physiological experiments when reversal of OIRD was achieved by both antagonists. Brains were quickly removed and flash frozen in liquid nitrogen. Trunk blood was collected into capillary collection tubes containing lithium heparin (Greiner Bio-One; Fisher Scientific). The blood was spun on a centrifuge for 10 min at 4°C, and plasma was carefully extracted from the supernatant. Brains and plasma samples underwent PK analyses for detection of NLX or NLXM at both time points. Standard rat PK studies dosed the candidate MOR antagonists and sample blood over multiple time points. A general description of tissue distribution can be obtained during PK modeling by calculating the volume of distribution. Samples underwent a direct measurement of peripheral restriction by measuring compound concentration in plasma and brain.

## Conditioned place aversion

The aversive effects following reversal of OIRD were measured using CPA (*Tzschentke, 2007*) using the video tracking system ANY-maze (RRID:SCR_014289). During a pre-conditioning session (Day 1), rats were allowed to explore both sides of a two compartment CPA box for 15 min. On

the Conditioning Day (Day 2), a divider was placed in the middle of the box to restrict movement to one side on the box. Rats were then randomly assigned to one side of the CPA box, and this side was paired with either saline (1 ml/kg), NLX (1 mg/kg), or NLXM (1 or 5 mg/kg). Rats were given intravenous fentanyl (50 μg/kg) or saline in their home cages. Four minutes later, they were placed in the paired side of the box and immediately given saline or an MOR antagonist. Rats were confined to this compartment for 15 min. For all CPA experiments, measures were taken to balance the side of CPA box by drug and by sex. On the post-conditioning test (Day 3), the divider was removed. Rats were then placed in the CPA box and allowed to explore both sides of the CPA box for 15 min. The total time spent in the paired chamber and CPA scores were calculated for all groups.

## Wireless in vivo fiber photometry

To selectively measure nTS neuronal calcium transient activity, we injected an AAV containing a calcium sensor expressed under the synapsin promoter (pGP-AAV9-syn-jGCaMP8m-WPRE; Addgene viral prep # 162375-AAV9) into the nTS (500 nl per side; 0.3–0.5 mm anterior, ±0.4 mm lateral, and 0.4 mm ventral to brain surface, relative to calamus scriptorius). Rats were allowed at least 3 weeks to recover from nanoinjection surgery to allow for robust viral expression in the nTS before IV catheters were implanted as described above. On the day of the experiment, rats were deeply anesthetized with isoflurane (3–4% induction, 2% maintenance), placed in the stereotaxic apparatus, and the nTS was exposed as described above. To overcome limitations associated with chronic fiber implants in the nTS, a wireless photometry headstage (TeleFipho, Amuza Inc) was secured to a 5 mm flat-tipped silica optic fiber cannula (400 μm c.d./470 μm o.d., NA 0.37, 2.5 mm receptacle; Doric) and mounted the stereotaxic arm with an apparatus 3D printed in house to permit simultaneous nTS exposure and photometry recordings. The optic fiber was lowered to the site of the dorsal brainstem and firmly pressed on the surface above the nTS until a signal was detected (~0.3–0.5 mm ventral from the surface). An LED generated blue light was bandpass-filtered (445–490 nm) to excite jGCaMP8m and emission fluorescence was detected by an internal photodiode detector for green light (bandpass filtered at 500–550 nm). An internal DC amplifier transmitted the data (16-bit arbitrary unit; AU) to a wireless receiver (TeleFipho, Amuza Inc) and the data was extracted in real-time using TeleFipho software (Amuza Inc) at a sampling rate of 100 Hz. The offset was set to 90° and the LED power was adjusted to achieve an optimal signal output (30–40 k a.u.) not to exceed 36 mW power, and remained constant throughout the recording. Baseline activity of GCaMP fluorescence nTS-infected cells was measured after at least 5 min to mitigate effects of baseline drift in signal due to slow photobleaching artifacts. Then, baseline activity was measured for at least 30 s prior to administration of either fentanyl (20 μg/kg, i.v.) or NLXM (5 mg/kg, i.v.). After response activity was recorded, rats received a second intravenous injection of either an opioid receptor antagonist (NLX,1 mg/kg or NLXM, 5 mg/kg) or fentanyl (20 μg/kg), respectively. Each intravenous administration was manually time locked to calcium transient recordings by a second experimenter and response activity following each injection was recorded for 60 s before administration of the second injection or conclusion of the experiment. Rats were transcardially perfused with PFA at the end of the experiment as described above and viral expression was validated for all animals.

For analysis of calcium transient activity, custom MATLAB scripts (https://doi.org/10.5281/zenodo.15042135) were used as previously described (*Higginbotham et al., 2025*). The raw data were fit to a double-exponential curve and subtracted from the raw data to account for any residual effects of baseline drift. MATLAB Signal Analyzer (RRID:SCR_001622) was used to extract regions of interest (ROIs) in the signal in 90 s traces (30 s pre-treatment, 60 s post-treatment) and traces were aligned to the time of injection. ROIs were standardized as z-scores using the sample standard deviation, $S$, where $S = \sqrt{\frac{\sum_{i=1}^{n}(x_i - \bar{X})^2}{n-1}}$ such that signals were centered to have a mean of 0 and scaled to have a standard deviation of 1. Based on this, the AUC of ROI z-scores were calculated in 30 s bins using the trapezoidal rule (*trapz* function in MATLAB) and one-sample t-tests were used to determine differences relative to baseline (theoretical mean = 0). For presentation, transients were down sampled by a factor of 10 with a phase offset of 2 and the resultant 10 Hz traces were smoothed using a moving mean duration of 5% of total ROI.

## Immunohistochemistry

Immunohistochemical procedures were performed as previously described (*Ruyle et al., 2019*). Brainstem sections containing the nTS were sliced using a cryostat (Leica) at a thickness of 40 μm. Free-floating sections were rinsed in 0.01 M PBS, blocked in 10% Normal Donkey Serum (Sigma) in 0.01 M PBS-Triton, and then incubated with the following antibodies: cFos (Rabbit anti-cFos; ab190289, Abcam; RRID:AB_2737414); tyrosine hydroxylase (Mouse anti-TH; MAB318, Millipore; RRID:AB_2201528). The following day, sections were rinsed with 0.01 M PBS, incubated with Donkey anti-Rabbit Cy3 and Donkey anti-Mouse Alexa fluor 488 (1:200; in 3% NDS and 0.3% Triton in 0.01 M PBS) and used to visualize targets of interest (Jackson ImmunoResearch).

## Image acquisition

IR and viral expression were examined with a Leica DMR microscope at ×20 magnification. 20 μm thick z-stacks of the nTS (2 μm per plane) were collected. ImageJ (v. 1.48, NIH) software was used for post-processing and analysis. In all groups, three sections of the nTS (corresponding to –360, 0, +360 μm relative to calamus scriptorius) were collected.

## Brainstem immunohistochemical analyses

Using ImageJ software (RRID:SCR_003070), the regions containing the nTS were outlined, and unilateral counts of Fos-IR, TH-IR, and co-labeled Fos- and TH-IR neurons were performed using a custom-made plugin (Cell Counter) for each of the 3 nTS sections per animal. Cell counts were performed by two individuals blinded to the treatment groups. The total number of each phenotype was summed from these three sections, and the percentage of TH-IR cells co-expressing Fos-IR was determined.

## Statistics

Statistical analyses were performed using GraphPad Prism v10.4.0. Data collection and analyses were performed blinded to the conditions of each experiment. For all experiments, the normality of sample data was determined using D'Agostino and Pearson tests and Shapiro–Wilk tests. Statistical significance was taken as *$p < 0.05$, **$p < 0.01$, ***$p < 0.001$, and ****$p < 0.0001$, as determined by two-way repeated measures ANOVA followed by Sidak's post hoc test; one-way ANOVA with Tukey post hoc test, two-tailed unpaired *t*-test, two-tailed paired *t*-test, and one-sample *t*-test. Detailed statistical reporting for all results is available in the Source data files.

# Acknowledgements

We thank Justin Meyer for breeding the rat colonies that were used in these experiments and for general managerial assistance throughout this project. We also thank Mike Cameron (Scripps Institute) for mass spectrometry analyses used for all biodistribution experiments. We also wish to thank Kristine Yoon and Thomas Schodl for their technical assistance. National Institutes of Health grant DA042499 (JAM), National Institutes of Health grant DA045463 (JAM), National Institutes of Health grant DA054900 (JAM), National Institutes of Health grant DA059067 (JAM), Washington University McDonnell Small Grants Program GF0012452 (BCR), and National Institutes of Health grant T32DA007261 (BCR).

# Additional information

## Funding

| Funder | Grant reference number | Author |
| --- | --- | --- |
| National Institutes of Health | DA042499 | Jose A Morón |
| National Institutes of Health | DA045463 | Jose A Morón |
| National Institutes of Health | DA054900 | Jose A Morón |

| Funder | Grant reference number | Author |
|---|---|---|
| National Institutes of Health | DA059067 | Jose A Morón |
| Washington University in St. Louis | GF0012452 | Brian C Ruyle |
| National Institutes of Health | T32DA007261 | Brian C Ruyle |

The funders had no role in study design, data collection, and interpretation, or the decision to submit the work for publication.

## Author contributions

Brian C Ruyle, Conceptualization, Data curation, Software, Formal analysis, Supervision, Funding acquisition, Validation, Investigation, Visualization, Methodology, Writing – original draft, Project administration, Writing – review and editing; Sarah Masud, Data curation, Formal analysis, Validation, Investigation, Project administration; Rohith Kesaraju, Data curation, Formal analysis, Investigation, Project administration, Writing – review and editing; Mubariz Tahirkheli, Data curation, Formal analysis, Project administration; Juhi Modh, Data curation, Formal analysis, Validation, Investigation, Project administration, Writing – review and editing; Caroline G Roth, Data curation, Project administration; Sofia Angulo-Lopera, Data curation, Formal analysis, Project administration, Writing – review and editing; Tania Lintz, Data curation, Validation, Investigation, Project administration, Writing – review and editing; Jessica A Higginbotham, Data curation, Software, Formal analysis, Validation, Project administration, Writing – review and editing; Nicolas Massaly, Conceptualization, Data curation, Methodology, Writing – review and editing; Jose A Morón, Conceptualization, Resources, Software, Supervision, Funding acquisition, Methodology, Writing – review and editing

## Author ORCIDs

Brian C Ruyle ⬥ https://orcid.org/0000-0001-8507-8218
Jose A Morón ⬥ https://orcid.org/0000-0003-3216-2488

## Ethics

All animals in this study were handled according to approved Institutional Animal Care and Use Committee (IACUC) protocol (#24-0068) of Washington University in St. Louis. All surgical and experimental procedures were approved by Washington University Committee in accordance with the National Institutes of Health Guidelines for the Care and Use of Laboratory Animals and Animal Research: Reporting In Vivo Experiments (ARRIVE) guidelines. Rats were initially group-housed with two to three animals per cage on a 12/12-hr dark/light cycle (lights on at 7:00) and acclimated to the animal facility holding rooms for at least 7 days before any manipulation. Following catheterization surgery, rats were single-housed and remained on an identical 12/12-hr dark/light cycle. The temperature for the holding rooms of all animals ranged from 21 to 24°C while the humidity was between 30 and 70%. Rats received food and water ad libitum for the duration of all experiments. All surgery was performed under isoflurane anesthesia, and every effort was made to minimize suffering.

Reviewer #1 (Public review): https://doi.org/10.7554/eLife.104469.3.sa1
Reviewer #2 (Public review): https://doi.org/10.7554/eLife.104469.3.sa2
Reviewer #3 (Public review): https://doi.org/10.7554/eLife.104469.3.sa3
Author response https://doi.org/10.7554/eLife.104469.3.sa4

---

# Additional files

## Supplementary files

Supplementary file 1. Tables of baseline, nadir and recovery of cardiorespiratory parameters in male and female rats that received intravenous administration of 20 or 50 μg/kg fentanyl. (A) Baseline cardiorespiratory values in rats prior to receiving 20 μg/kg fentanyl. Values are mean ± SE. Data are from rats used in *Figures 1, 3 and 4* and are separated by sex. There was no significant difference in any of the measured parameters between male and female rats. Oxygen saturation:

one-way ANOVA, $F_{(5,27)}$ = 0.6787, p = 0.6424; heart rate: one-way ANOVA, $F_{(5,27)}$ = 1.078, p = 0.3944; respiratory rate: one-way ANOVA, $F_{(5,27)}$ = 1.527, p = 0.2147. (B) Baseline cardiorespiratory values in male and female rats prior to receiving 50 µg/kg fentanyl. Values are mean ± SE. Data are from rats used in *Figure 1*, *Figure 3—figure supplement 1*, and *Figure 4—figure supplement 1* and are separated by sex. Female rats in *Figure 1* had a significantly higher heart rate at baseline compared male rats in *Figure 1*. Oxygen saturation: one-way ANOVA, $F_{(5,25)}$ = 1.161, p = 0.3558; heart rate: one-way ANOVA, $F_{(5,25)}$ = 3.774, p = 0.0110, Tukey's post hoc test [†]p < 0.05 *Figure 1* males versus *Figure 1* females; respiratory rate: one-way ANOVA, $F_{(5,25)}$ = 1.728, p = 0.1649. (C) Nadir and recovery values in male and female rats that received 20 µg/kg fentanyl. Values are mean ± SE. Data are from rats used in *Figures 1, 3, and 4* and are separated by sex. For nadir data: oxygen saturation: one-way ANOVA, $F_{(5,27)}$ = 1.229, p = 0.3230; heart rate: one-way ANOVA, $F_{(5,27)}$ = 1.844, p = 0.1379; respiratory rate: one-way ANOVA, $F_{(5,27)}$ = 2.669, p = 0.0438. Despite the significant interaction for respiratory rate, no post hoc differences were detected. Sex differences in recovery times were also evaluated. oxygen saturation: one-way ANOVA, $F_{(5,27)}$ = 0.7021, p = 0.6267; heart rate: one-way ANOVA, $F_{(5,27)}$ = 0.4536, p = 0.8069; respiratory rate: one-way ANOVA, $F_{(5,27)}$ = 1.100, p = 0.3832. (D) Nadir and recovery values in male and female rats that received 50 µg/kg fentanyl. Values are mean ± SE. Data are from rats used in *Figure 1*, *Figure 3—figure supplement 1*, and *Figure 4—figure supplement 1* and are separated by sex. For nadir data: oxygen saturation: one-way ANOVA, $F_{(5,25)}$ = 1.384, p = 0.2639; heart rate: one-way ANOVA, $F_{(5,25)}$ = 0.8879, p = 0.5039; respiratory rate: one-way ANOVA, $F_{(5,25)}$ = 1.154, p = 0.3591. Sex differences in recovery times were also evaluated. oxygen saturation: one-way ANOVA, $F_{(5,25)}$ = 0.2277, p = 0.9469; heart rate: one-way ANOVA, $F_{(5,25)}$ = 1.591, p = 0.1992; respiratory rate: one-way ANOVA, $F_{(5,25)}$ = 4.258, p = 0.0061. Tukey's post hoc test [†]p < 0.05 *Figure 1* males versus *Figure 1* females, [#]p < 0.05 *Figure 1* females versus *Figure 4—figure supplement 1* males.

MDAR checklist

Source data 1. Excel spreadsheeet of statistical source data for *Supplementary file 1*.

### Data availability

The authors confirm that all relevant data reported in this study are provided in the supplementary and source data files. All original code for MATLAB analysis workflow is available through https://doi.org/10.5281/zenodo.15042135. Further information and requests for resources and reagents should be directed to and will be fulfilled by the Lead Contact, Jose A. Morón (jmoron-concepcion@wustl.edu).

The following dataset was generated:

| Author(s) | Year | Dataset title | Dataset URL | Database and Identifier |
|---|---|---|---|---|
| Higginbotham JA | 2025 | Ruyle_Photometry | https://doi.org/10.5281/zenodo.15042135 | Zenodo, 10.5281/zenodo.15042135 |

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
