## [Editor Report · eLife Assessment]

This manuscript represents a **fundamental** contribution demonstrating that fentanyl-induced respiratory depression can be reversed with a peripherally-restricted mu opioid receptor antagonist. The paper reports **compelling** and rigorous physiological, pharmacokinetic, and behavioral evidence supporting this major claim, and furthers mechanistic understanding of how peripheral opioid receptors contribute to respiratory depression. These findings reshape our understanding of opioid-related effects on respiration and have significant therapeutic implications given that medications currently used to reverse opioid overdose (such as naloxone) produce severe aversive and withdrawal effects via actions within the central nervous system.

---

## [Referee Report · Reviewer #1 (Public review)]

Summary:

This paper shows that the synthetic opioid fentanyl induces respiratory depression in rodents. This effect is revised by the opioid receptor antagonist naloxone, as expected. Unexpectedly, the peripherally restricted opioid receptor antagonist naloxone methiodide also blocks fentanyl-induced respiratory depression.

Strengths:

The paper reports compelling physiology data supporting the induction of respiratory distress in fentanyl-treated animals. Evidence suggesting that naloxone methiodide reverses this respiratory depression is compelling. This is further supported by pharmacokinetic data suggesting that naloxone methiodide does not penetrate into the brain, nor is it metabolized into brain-penetrant naloxone.

Weaknesses:

The paper would be further strengthened by establishing the functional significance of the altered neural activity detected in the nTS (as measured by cFos and GcAMP/photometry) in the context of opioid-induced respiratory depression.

---

## [Referee Report · Reviewer #2 (Public review)]

Summary:

In this article, Ruyle and colleagues assessed the contribution of central and peripheral mu opioid receptors in mediating fentanyl-induced respiratory depression using both nalaxone and nalaxone methiodide, which does not cross the blood brain barrier. Both compounds prevented and reversed fentanyl-induced respiratory depression to a comparable degree. The advantage of peripheral treatments is that they circumvent the withdrawal-like effects of nalaxone. Moreover, neurons located in the nucleus of the solitary tract are no longer activated by fentanyl when nalaxone methiodide is administered, suggesting that these responses are mediated by peripheral mu opioid receptors. The results delineate a role for peripheral mu opioid receptors in fentanyl-derived respiratory depression and identify a potentially advantageous approach to treating overdoses without inflicting withdrawal on the patients.

Strengths:

The strengths of the article include the intravenous delivery of all compounds, which increases the translational value of the article. The authors address both prevention and reversal of fentanyl-derived respiratory depression. The experimental design and data interpretation are rigorous and appropriate controls were used in the study. Multiple doses were screened in the study and the approaches were multipronged. The authors demonstrated activation of NTS cells using multiple techniques and the study links peripheral activation of mu opioid receptors to central activation of NTS cells. Both males and females were used in the experiments. The authors demonstrate the peripheral restriction of nalaxone methiodide.

Weaknesses:

Nalaxone is already broadly used to prevent overdoses from opioids so in some respects, the effects reported here are somewhat incremental.

Comments on the latest version:

I think the authors have adequately addressed previous critiques and I don't have any additional comments.

---

## [Referee Report · Reviewer #3 (Public review)]

Summary

This manuscript outlines a series of very exciting and game-changing experiments examining the role of peripheral MORs in OIRD. The authors outline experiments that demonstrate a peripherally restricted MOR antagonist (NLX Methiodide) can rescue fentanyl-induced respiratory depression and this effect coincides with a lack of conditioned place aversion. This approach would be a massive boon to the OUD community, as there are a multitude of clinical reports showing that naloxone rescue post fentanyl over-intoxication is more aversive than the potential loss-of-life to the individuals involved. This important study reframes our understanding of successful overdose rescue with a potential for reduced aversive withdrawal effects.

Strengths:

Strengths include the plethora of approaches arriving at the same general conclusion, the inclusion of both sexes, and the result that a peripheral approach for OIRD rescue may side-step severe negative withdrawal symptoms of traditional NLX rescue.

Weaknesses:

All weaknesses were addressed.

---

## [Author Response]

The following is the authors’ response to the original reviews.

**eLife Assessment**
This manuscript represents a fundamental contribution demonstrating that fentanyl-induced respiratory depression can be reversed with a peripherally-restricted mu opioid receptor antagonist. The paper reports compelling and rigorous physiological, pharmacokinetic, and behavioral evidence supporting this major claim, and furthers mechanistic understanding of how peripheral opioid receptors contribute to respiratory depression. These findings reshape our understanding of opioid-related effects on respiration and have significant therapeutic implications given that medications currently used to reverse opioid overdose (such as naloxone) produce severe aversive and withdrawal effects via actions within the central nervous system.

We thank the reviewers for their insightful comments and critiques, which we have incorporated into the manuscript. We believe these revisions have significantly improved the manuscript. Additionally, following discussions among the authors, we have revised the color scheme across all figures. For example, the color of the symbols in Figure 1B-D now match the bars in Figure 1E-J, rather than the symbols. We feel that this change improves the clarity and visual consistency of the figures, making it easier to interpret the data across figures.

**Public Reviews:**

**Reviewer #1 (Public review):**
Summary:This paper shows that the synthetic opioid fentanyl induces respiratory depression in rodents. This effect is revised by the opioid receptor antagonist naloxone, as expected. Unexpectedly, the peripherally restricted opioid receptor antagonist naloxone methiodide also blocks fentanyl-induced respiratory depression.Strengths:The paper reports compelling physiology data supporting the induction of respiratory distress in fentanyl-treated animals. Evidence suggesting that naloxone methiodide reverses this respiratory depression is compelling. This is further supported by pharmacokinetic data suggesting that naloxone methiodide does not penetrate into the brain, nor is it metabolized into brain-penetrant naloxone.Weaknesses:A weakness of the study is the fact that the functional significance of opioid-induced changes in neural activity in the nTS (as measured by cFos and GcAMP/photometry) is not established. Does the nTS regulate fentanyl-induced respiratory depression, and are changes in nTS activity induced by naloxone and naloxone methiodide relevant to their ability to reverse respiratory depression?
**Reviewer #2 (Public review):**
Summary:In this article, Ruyle and colleagues assessed the contribution of central and peripheral mu opioid receptors in mediating fentanyl-induced respiratory depression using both naloxone and naloxone methiodide, which does not cross the blood-brain barrier. Both compounds prevented and reversed fentanyl-induced respiratory depression to a comparable degree. The advantage of peripheral treatments is that they circumvent the withdrawal-like effects of naloxone. Moreover, neurons located in the nucleus of the solitary tract are no longer activated by fentanyl when nalaxone methiodide is administered, suggesting that these responses are mediated by peripheral mu opioid receptors. The results delineate a role for peripheral mu opioid receptors in fentanyl-derived respiratory depression and identify a potentially advantageous approach to treating overdoses without inflicting withdrawal on the patients.Strengths:The strengths of the article include the intravenous delivery of all compounds, which increase the translational value of the article. The authors address both the prevention and reversal of fentanyl-derived respiratory depression. The experimental design and data interpretation are rigorous and appropriate controls were used in the study. Multiple doses were screened in the study and the approaches were multipronged. The authors demonstrated the activation of NTS cells using multiple techniques and the study links peripheral activation of mu opioid receptors to central activation of NTS cells. Both males and females were used in the experiments. The authors demonstrate the peripheral restriction of naloxone methiodide.Weaknesses:Nalaxone is already broadly used to prevent overdoses from opioids so in some respects, the effects reported here are somewhat incremental.

The reviewer is correct that naloxone is the standard antidote for reversing opioid-induced respiratory depression. However, its limitations, including the risk of precipitated withdrawal, are well-documented in both preclinical and clinical studies. The likelihood of withdrawal increases when multiple doses of naloxone are administered. Since naloxone-induced withdrawal is centrally mediated, this study aimed to evaluate a peripherally restricted MOR antagonist for its ability to prevent or reverse fentanyl-induced respiratory depression. A key finding is that NLXM reversed OIRD without inducing aversive behavior. This suggests that peripheral antagonists like NLXM may be integrated into intervention strategies that save lives while preventing the adverse behavioral and physiological effects that are observed after treatment with naloxone.

**Reviewer #3 (Public review):**
Summary:This manuscript outlines a series of very exciting and game-changing experiments examining the role of peripheral MORs in OIRD. The authors outline experiments that demonstrate a peripherally restricted MOR antagonist (NLX Methiodide) can rescue fentanyl-induced respiratory depression and this effect coincides with a lack of conditioned place aversion. This approach would be a massive boon to the OUD community, as there are a multitude of clinical reports showing that naloxone rescue post fentanyl over-intoxication is more aversive than the potential loss-of-life to the individuals involved. This important study reframes our understanding of successful overdose rescue with potential for reduced aversive withdrawal effects.Strengths:Strengths include the plethora of approaches arriving at the same general conclusion, the inclusion of both sexes and the result that a peripheral approach for OIRD rescue may side-step severe negative withdrawal symptoms of traditional NLX rescue.Weaknesses:The major weakness of this version relates to the data analysis assessed sex-specific contributors to the results.
**Recommendations for the authors:**

**Reviewer #1 (Recommendations for the authors):**
Some points for the authors to consider are:(1) In the Abstract, it is unclear why "high potency and lipophilicity" contribute to opioid-induced respiratory depression.

The higher potency of fentanyl compared to other opioids significantly increases the risk of overdose and subsequent respiratory depression. Its high lipophilicity facilitates rapid absorption and central nervous system penetration, which contributes to the rapid onset of these cardiorespiratory depression. The narrow therapeutic window of fentanyl further emphasizes the critical need for timely intervention when an overdose has occurred, and effective antagonists to reverse respiratory depression and save lives. We have revised the abstract to clarify these points.

(2) Are the doses of fentanyl used in the study (2, 20, or 50 µg/kg IV) relevant to those achieved by fentanyl-exposed human drug users?

In these studies, we intravenously administered three doses of fentanyl. The human equivalent doses (HED) of 20ug/kg and 50 ug/kg fentanyl are ~3 ug/kg and ~8 ug/kg, respectively. These doses have previously been shown to induce respiratory depression in humans (Dahan et al.,2005).

(3) In Figure 1, it appeared that only a small fraction of tyrosine hydroxylase-positive (TH+) neurons expressed cFos in response to fentanyl, and the degree of cFos expression was largely similar across all fentanyl doses tested. Thus, it is unclear whether TH+ neurons play a role in fentanyl-induced respiratory depression, and the value of these data is unclear (see point #6 below also).

As shown in the mean data, the lowest dose of fentanyl, which was below the threshold for inducing OIRD, activated approximately 50% of tyrosine hydroxylase-positive (TH+) nTS neurons. In contrast, the highest dose of fentanyl resulted in a statistically significant increase, with ~75% of TH+ cells co-expressing Fos-IR.

We included the assessment of catecholaminergic nTS cells for several reasons. The regions of the nTS evaluated in this study contains high expression of MOR and are the termination points of sensory afferent fibers transmitting cardiorespiratory information to the nTS (Aicher et al., 2000; Furdui et al., 2024). Catecholaminergic cells receive direct excitatory inputs from visceral afferents (Appleyard et al., 2007) and exhibit intensity-dependent increases in Fos-IR in rats exposed to hypoxic air (Kline et al., 2010; King et al., 2012). These neurons are essential for generating appropriate cardiorespiratory responses to hypoxic challenges (Bathina et al., 2013; King et al., 2015). As the reviewer notes, rats exposed to fentanyl exhibit a high degree of Fos-IR in the nTS, including catecholaminergic neurons. Despite the robust fentanyl-induced activation (increased Fos-IR) nTS neurons, yet there appears to be a failure to initiate appropriate chemoreflex-mediated cardiorespiratory responses. Our photometry data further indicate that fentanyl-induced changes in neuronal activity are mediated, in part, by peripheral MOR. Collectively, these findings suggest that fentanyl impacts nTS activity through alterations in peripheral afferent signaling to the nTS, which may contribute to the severity and duration of OIRD.

(4) It would help with the flow of the paper if the pharmacokinetic data shown in Figure 6 were presented earlier (as part of Figure 2).

We have moved the biodistribution data earlier in the manuscript, now presenting it as Figure 2. The numbering of all subsequent figures has been adjusted accordingly.

(5) In Figure 5, there appears to be a large number of GCaMP-expressing neurons located outside the nTS. To what degree can the changes in calcium signaling, attributed to alterations in neural activity in the nTS, be explained by altered activity of neurons located outside the nTS?

The reviewer is correct that our viral spread extends beyond the boundaries of the nTS, raising the possibility that the responses observed in Figure 5 may be influenced by neural activity of cells outside the nTS. While some viral spread beyond the target region is unavoidable, calcium transients were measured at the tip of the fiber, which was positioned directly within the nTS.

To address this concern further, we performed Fos immunohistochemistry in a subset of animals that received bilateral GCaMP virus injections into the nTS. Following fentanyl administration (50 µg/kg IV), brains were collected two hours later. As shown in the accompanying image, we observed Fos-IR co-expression with GCaMP exclusively within the nTS boundaries. No Fos-IR was detected outside the nTS, including in GCaMP cells. Taken together, these findings support our conclusion that the data depicted in our photometry figure (now Figure 6) accurately represent fentanyl-induced activity changes in nTS neurons.

**Author response image 1. sa4fig1:** Arrowheads: Fos-negative GCaMP cell; Arrows: Co-labeled Fos/GCaMP cell; Asterisk: Fos+ GCaMP-negative cell.

(6) Currently, the cFos and photometry data are descriptive in nature. Are opioid-induced changes in nTS neural activity relevant to respiratory depression? If so, one might expect DREADD-mediated stimulation of the nTS neural activity (or stimulating nTS activity by some other means) would reverse fentanyl-induced respiratory depression similar to naloxone and methyl-naloxone.

The reviewer raises an interesting point regarding the relevance of the nTS in the context of OIRD. The nTS is a major site of integration of sensory afferent information and involved in the initiation of reflex responses that facilitate a return to homeostasis. As described above, we characterized the collective response of nTS neurons to intravenous fentanyl using both Fos immunohistochemistry and fiber photometry. Our data indicate that fentanyl-induced changes in nTS activity are strongly mediated by peripheral MOR. While the suggestion to use global chemogenetic activation of nTS neurons to reverse fentanyl-induced respiratory depression is intriguing, results from these experiments may be difficult to interpret due to the extensive heterogeneity of the nTS. However, we are currently conducting similar experiments using a more selective approach that will allow us to isolate and evaluate specific nTS phenotypes to better understand their contributions to OIRD.

(7) Are peripherally restricted mu opioid receptor (MOR) agonists available? If so, it would strengthen the paper if such compounds could be used to show that stimulation of peripheral MORs is sufficient to induce respiratory distress independent of actions on centrally located MORs.

Peripherally acting Mu Opioid Receptor Antagonists (PAMORAs) are indeed available and currently being evaluated in our laboratory.

**Reviewer #2 (Recommendations for the authors):**
Consider having the figures/data numbered in the order that they appear in the manuscript. Right now, Figure 6 is mentioned between Figures 1 and 2 (minor).

Thank you for this suggestion. We have reordered the figures so that the biodistribution figure appears before the MOR antagonist pretreatment and reversal figures.

**Reviewer #3 (Recommendations for the authors):**
This manuscript outlines a series of very exciting and game-changing experiments examining the role of peripheral MORs in OIRD. The authors outline experiments that demonstrate a peripherally restricted MOR antagonist (NLX Methiodide) can rescue fentanyl-induced respiratory depression and this effect coincides with a lack of conditioned place aversion. This approach would be a massive boon to the OUD community, as there are a multitude of clinical reports showing that naloxone rescue post fentanyl over-intoxication is more aversive than the potential loss-of-life to the individuals involved. This important study reframes our understanding of successful overdose rescue with potential for reduced aversive withdrawal effects.While this is an exciting and important study, there are a few minor to moderate critiques for the authors to consider. These are below.(1) Title: "devoid of aversive effects" - While CPA is a good, cumulative indicator of potential aversive effects, it is not an exhaustive one. Since no other withdrawal measures were included, this is an overstatement.

The reviewer is correct in noting that our analysis of aversive effects is not exhaustive. Since we only assessed changes in aversive behavior between NLX and NLXM, we believe it is more accurate to modify the title accordingly. We have changed the title from “devoid of aversive effects” to “devoid of aversive behavior” better reflect the scope of the experiments conducted.

(2) Page 3, top line: MOR (mu opioid receptor) is highly expressed...An article should likely be included prior to MOR or make plural and adjust the sentence.

Thank you for this suggestion. We have reworked this section in the manuscript.

(3) Figure 6D: this figure is very important for the interpretation of every single figure. It should either be moved to figure 1 or 2 or combined with figure 1 or 2.

Thank you for this suggestion. The biodistribution figure has been moved to Figure 2.

(4) Page 5, line 164, Figure 21-D: remove the 1.

Done.

(5) Sex differences (or lack thereof):Throughout the manuscript, the authors report a lack of sex differences. However, while the data is not powered for the distinction of sex differences, there appears to be a bi-modal distribution of the individual data points that likely correspond to sex across most experiments. For example, in Figure 2E there are both color and clear dots, which this reviewer assumes indicates sex (however, this wasn't easily apparent if it was commented on at all in the paper). If you look at the saline oxygen saturation (nadir) levels (2e), there is wide variability with the red-filled circles, but not the clear ones. This may indicate a bimodal distribution (and may be related to the baseline HR sex differences highlighted). This is also the case in Figure 2L but is perhaps more obvious in the CPA score data (Figure 4d), where it seems the nlx negative CPA effects were likely driven primarily by one sex. While this reviewer does not expect a full powering of experiments for sex differences (and also is very appreciative of the inclusion of both sexes), full raw data with sex indicated included in the supplemental data would greatly aid the field in general and allow for those with a specific interest in this area to build upon this data. Additionally, further discussion regarding the potential role of sex differences in the translational value of these findings is also warranted.

For all bar graphs, open symbols represent females and filled symbols represent males. This information can be found in the first paragraph of the Materials and Methods section. We have also added this information to each figure for increased visibility. We appreciate the acknowledgement of our inclusion of both sexes. For all experiments, we attempted to balance by sex. Unfortunately, we occasionally had to exclude animals for technical reasons (with clogged catheters being the most common reason for exclusion). This sometimes led to an imbalance in sex in some groups, as the reviewer has noted. In the graph of oxygen saturation nadir values in Fig 2E now Fig 3E in the revised manuscript, all animals received intravenous fentanyl at a dose of 20 ug/kg. The reviewer is correct that there is greater variability in the males (filled symbols) compared to the females (open symbols) in this graph. However, this variability in the distribution was not observed in Fig 1E or Fig 4E, in which male and female rats received an identical dose of 20 ug/kg. Taking this into account, our overall interpretation of the data is that there is relatively minor sex difference in the responses observed after intravenous fentanyl, and the variability in Fig 3E is primarily due to a lower n compared to Fig 1E.

All raw data will be uploaded to a data repository.

(6) Page 7, line 209: Figure 5D should be Figure 6D.

We have incorporated this change.

(7) Page 8, line 267: Cure should be Curve.

We have incorporated this change.

(8) Discussion: Page10, line322 states that "no detectable NLX ... was found in brain tissue". This is incorrect based on Figure 6.

The sentence the reviewer highlighted refers to detection of NLX or NLXM in brain tissue from animals that received intravenous NLXM. As demonstrated in the biodistribution figure (now Figure 2 in the manuscript), our data demonstrate that an intravenous injection of NLXM did not result in NLX formation in the brain. We have reworked the sentence for clarity.

(9) jGCaMP injections: Figure 5B/c shows the distribution of the gcamp across animals. The optic fiber is placed directly over the NTs. However, how are we certain there isn't a nearby nuclei/structure outside the NTS that is contributing to the photometry data presented in D-G?

See our above comment.

(10) Fiber Photometry and Sex: These studies unfortunately may have had only 1 of a sex included in the fiber photometry data. While the inclusion is overall good, the single value for a sex suggests that there are differences, given the clustering of the data. While the anesthesia may be driving this potential sex effect, it is not clear based on the data presented. For reference: https://link.springer.com/article/10.1007/s12975-012-0229-y

The reviewer is correct that there was an imbalance of sex in this dataset. While we made every attempt to balance for sex across all experiments, we unfortunately had to exclude some animals for technical reasons (clogged catheter, missed injection site, etc). This produced an imbalance in our photometry studies and did not allow us to thoroughly evaluate sex differences in fentanyl-induced changes in neural activity or in the responses to anesthesia. We have expanded on this limitation in the discussion.

(11) Figure 5 - the bars are not the color indicated by the legend.

We have corrected this in the figure. Thank you.